# Online Convex Matrix Factorization with Representative Regions

**Abhishek Agarwal** *
Electrical and Computer Engineering
University of Illinois Urbana-Champaign
abhiag@illinois.edu

**Jianhao Peng** *
Electrical and Computer Engineering
University of Illinois Urbana-Champaign
jianhao2@illinois.edu

**Olgica Milenkovic**
Electrical and Computer Engineering
University of Illinois Urbana-Champaign
milenkov@illinois.edu

## Abstract

Matrix factorization (MF) is a versatile learning method that has found wide applications in various data-driven disciplines. Still, many MF algorithms do not adequately scale with the size of available datasets and/or lack interpretability. To improve the computational efficiency of the method, an online (streaming) MF algorithm was proposed in [1]. To enable data interpretability, a constrained version of MF, termed convex MF, was introduced in [2]. In the latter work, the basis vectors are required to lie in the convex hull of the data samples, thereby ensuring that every basis can be interpreted as a weighted combination of data samples. No current algorithmic solutions for online convex MF are known as it is challenging to find adequate convex bases without having access to the complete dataset. We address both problems by proposing the first online convex MF algorithm that maintains a collection of *constant-size* sets of representative data samples needed for interpreting each of the basis [2] and has the same almost sure convergence guarantees as the online learning algorithm of [1]. Our proof techniques combine random coordinate descent algorithms with specialized quasi-martingale convergence analysis. Experiments on synthetic and real world datasets show significant computational savings of the proposed online convex MF method compared to classical convex MF. Since the proposed method maintains small representative sets of data samples needed for convex interpretations, it is related to a body of work in theoretical computer science, pertaining to generating point sets [3], and in computer vision, pertaining to archetypal analysis [4]. Nevertheless, it differs from these lines of work both in terms of the objective and algorithmic implementations.

## 1 Introduction

Matrix Factorization (MF) is a widely used dimensionality reduction technique [5, 6] whose goal is to find a basis that allows for a sparse representation of the underlying data [7, 8]. Compared to other dimensionality reduction techniques based on eigendecompositions [9], MF enforces fewer restrictions on the choice of the basis and hence ensures larger representation flexibility for complex datasets. At the same time, it provides a natural, application-specific interpretation for the bases.

MF methods have been studied under various modeling constraints [2, 10, 11, 12, 13, 14, 15, 16]. The most frequently used constraints are non-negativity, constraints that accelerate convergence rates, semi-non-negativity, orthogonality and convexity [11, 2, 17]. Convex MF (cvxMF) [2] is of special interest as it requires the basis vectors to be convex combinations of the observed data samples [18, 19]. This constraint allows one to interpret the basis vectors as probabilistic sums of a (small) representative subsets of data samples.

Unfortunately, most of the aforementioned constrained MF problems are non-convex and NP-hard [20, 21, 22], but can often be suboptimally solved using alternating optimization approaches for finding local optima [13]. Alternating optimization approaches have scalability issues since the number of matrix multiplications and convex optimization steps in each iteration depends both on the data set size and its dimensionality. To address the scalability issue [23, 24, 25], Mairal, Bach, Ponce and Sapiro [1] introduced an online MF algorithm that minimizes a surrogate function amenable to sequential optimization. The online algorithm comes with strong performance guarantees, asserting that its solution converges almost surely to a local optima of the generalization loss.

Currently, no online/streaming solutions for convex MF are known as it appears hard to satisfy the convexity constraint without having access to the whole dataset. We propose the first online MF method accounting for convexity constraints on *multi-cluster data sets*, termed online convex Matrix Factorization (**online cvxMF**). The proposed method solves the cvxMF problem of Ding, Li and Jordan [2] in an online/streaming fashion, and allows for selecting a *collection* of "typical" representative sets of individual clusters (see Figure 1). The method sequentially processes single data sample and updates a running version of a collection of constant-size sets of representative samples of the clusters, needed for convex interpretations of each basis element. In this case, the basis also plays the role of the cluster centroid, and further increases interpretability. The method also allows for both sparse data and sparse basis representations. In the latter context, sparsity refers to restricting each basis to be a convex combination of data samples in a small representative region. The online cvxMF algorithm has the same theoretical convergence guarantees as [1].

We also consider a more restricted version of the cvxMF problem, in which the representative samples are required to be strictly contained within their corresponding clusters. The algorithm is semi-heuristic as it has provable convergence guarantees only when sample classification is error-free, as is the case for non-trivial supervised MF [26] (note that applying [1] to each cluster individually is clearly suboptimal, as one needs to jointly optimize both the basis and the embedding). The restricted cvxMF method nevertheless offers excellent empirical performance when properly initialized.

It is worth pointing out that our results complement a large body of work that generalize the method of [1] for different loss functions [27, 28, 29] but do not impose convexity constraints. Furthermore, the proposed online cvxMF exhibits certain similarities with online generating point set methods [3] and online archetypal analysis [4]. The goal of these two lines of work is to find a small set of representative samples whose convex hull contains the *majority* of observed samples. In contrast, we only seek a small set of representative samples needed for *accurately describing a basis of the data*.

The paper is organized as follows. Section 2 introduces the problem, relevant notation and introduces our approach towards an online algorithm for the cvxMF problem. Section 3 describes the proposed online algorithm and Section 4 establishes that the learned basis almost surely converge to a stationary point of the approximation-error function. The theoretical guarantees hold under mild assumptions on the data distribution reminiscent of those used in [1], while the proof techniques combine *random coordinate descent algorithms with specialized quasi-martingale convergence analysis*. The performance of the algorithm is tested on both synthetic and real world datasets, as outlined in Section 5. The real world datasets include are taken from the UCI Machine Learning [30] and the 10X Genomics repository [31]. The experiments reveal that our online cvxMF runs four times faster than its non-online counterpart on datasets with $10^4$ samples, while for larger sample sets cvxMF becomes exponentially harder to execute. The online cvxMF also produces high-accuracy clustering results.

## 2 Notation and Problem Formulation

We denote sets by $[l] = \{1, \ldots, l\}$. Capital letters are reserved for matrices (bold font) and random variables (RVs) (regular font). Random vectors are described by capital underlined letters, while deterministic vectors are denoted by lower-case underlined letters. We use $\mathbf{M}[l]$ to denote the $l^{\text{th}}$ column of the matrix $\mathbf{M}$, $\mathbf{M}[r, l]$ to denote the element in row $r$ and column $l$, and $\underline{x}[l]$ to denote the

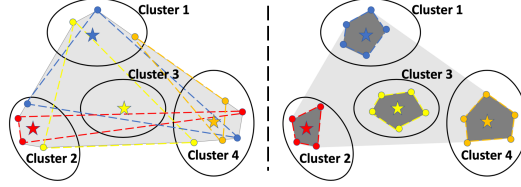

Figure 1: A multi-cluster dataset: Stars represent the learned bases, while circles denote representative samples for the basis of the same color. Left: The representative sets for the individual basis elements are unrestricted. Right: The representative sets are restricted to lie within their corresponding clusters.

$l^{\text{th}}$ coordinate of a vector $\underline{x}$. Furthermore, col($\mathbf{M}$) stands for the set of columns of $\mathbf{M}$, while cvx($\mathbf{M}$) stands for the convex hull of col($\mathbf{M}$).

Let $\mathbf{X} \in \mathbb{R}^{m \times n}$ denote a matrix of $n$ data samples of constant dimension $m$ arranged (summarized) column-wise, let $\mathbf{D} \in \mathbb{R}^{m \times k}$ denote the $k$ basis vectors used to represent the data and let $\mathbf{\Lambda} \in \mathbb{R}^{\mathbf{k} \times \mathbf{n}}$ stand for the low-dimension embedding matrix. The classical MF problem reads as:

$$\min_{\mathbf{D}, \mathbf{\Lambda}} \|\mathbf{X} - \mathbf{D}\mathbf{\Lambda}\|_2^2 + \lambda \|\mathbf{\Lambda}\|_1. \tag{1}$$

where $\|\underline{x}\|_2 \triangleq \sqrt{\underline{x}^T \underline{x}}$ and $\|\underline{x}\|_1 \triangleq \sum_j |\underline{x}[j]|$ denote the $\ell_2$-norm and $\ell_1$-norm of the vector $\underline{x}$, respectively.

In practice, $\mathbf{X}$ is inherently random and in the stochastic setting it is more adequate to minimize the above objective in expectation. In this case, the data approximation-error $g(\mathbf{D})$ for a fixed $\mathbf{D}$ equals:

$$g(\mathbf{D}) \triangleq \mathbb{E}_{\underline{X}}[\min_{\underline{\alpha} \in \mathbb{R}^k} \|\underline{X} - \mathbf{D}\underline{\alpha}\|_2^2 + \lambda \|\underline{\alpha}\|_1], \tag{2}$$

where $\underline{X}$ is a random vector of dimension $m$ and the parameter $\lambda$ controls the sparsity of the coefficient vector $\underline{\alpha}$. For analytical tractability, we assume that $\underline{X}$ is drawn from the union of $k$ disjoint, convex compact regions (clusters), $C^{(i)} \in \mathbb{R}^m$, $i \in [k]$. Each cluster is independently selected based on a given distribution, and the vector $\underline{X}$ is sampled from the chosen cluster. Both the cluster and intra-cluster sample distributions are mildly constrained, as described in the next section.

The approximation-error of a single data sample $\underline{x} \in \mathbb{R}^m$ with respect to $\mathbf{D}$ equals

$$\ell(\underline{x}, \mathbf{D}) \triangleq \min_{\underline{\alpha} \in \mathbb{R}^k} \frac{1}{2} \|\underline{x} - \mathbf{D}\underline{\alpha}\|_2^2 + \lambda \|\underline{\alpha}\|_1. \tag{3}$$

Consequently, the approximation error-function $g(\mathbf{D})$ in Equation (2) may be written as $g(\mathbf{D}) = \mathbb{E}_{\underline{X}}\big[\ell(\underline{X}, \mathbf{D})\big]$. The function $g(\mathbf{D})$ is non-convex and optimizing it is NP-hard and requires prior knowledge of the distribution. To mitigate the latter problem, one can revert to an empirical estimate of $g(\mathbf{D})$ involving the data samples $\underline{x}_n$, $n \in [t]$,

$$g_t(\mathbf{D}) = \frac{1}{t} \sum_{n=1}^{t} \ell(\underline{x}_n, \mathbf{D}).$$

Maintaining a running estimate of $\mathbf{D}_t$ of an optimizer of $g_t(\mathbf{D})$ involves updating the coefficient vectors for all the data samples observed up to time $t$. Hence, it is desirable to use *surrogate functions* to simplify the updates. The surrogate function $\hat{g}_t(\mathbf{D})$ proposed in [1] reads as

$$\hat{g}_t(\mathbf{D}) \triangleq \frac{1}{t} \sum_{n=1}^{t} \frac{1}{2} \|\underline{x}_n - \mathbf{D}\underline{\alpha}_n\|_2^2 + \lambda \|\underline{\alpha}_n\|_1, \tag{4}$$

where $\underline{\alpha}_n$ is an approximation of the optimal value of $\underline{\alpha}$ at step $n$, computed by solving Equation (3) with $\mathbf{D}$ fixed to $\mathbf{D}_{n-1}$, an optimizer of $\hat{g}_{n-1}(\mathbf{D})$.

The above approach lends itself to an implementation of an online MF algorithm, as the sum in Equation (4) may be efficiently optimized whenever adding a new sample. However, in order to satisfy the

convexity constraint of [2], all previous values of $\underline{x}_n$ are needed to update $\mathbf{D}$. To mitigate this problem, we introduce for each cluster $C^{(i)}$ a representative set $\hat{\mathbf{X}}_t^{(i)} \in \mathbb{R}^{m \times N_i}$ and its convex hull (representative region) $\mathrm{cvx}(\hat{\mathbf{X}}_t^{(i)})$. The values of $N_i$ are kept constant, and we require $\mathbf{D}_t[i] \in \mathrm{cvx}(\hat{\mathbf{X}}_t^{(i)})$. As illustrated in Figure 1, we may further restrict the representative regions as follows.

$\mathcal{R}_u$ (Figure 1, Left): We only require that $\hat{\mathbf{X}}_t^{(i)} \subset \mathrm{cvx}(\bigcup_j C^{(j)}), i \in [k]$. This unrestricted case leads to an online solution for the cvxMF problem [2] as one may use $\bigcup_j \hat{\mathbf{X}}_t^{(j)}$ as a single representative region. The underlying online algorithms has provable performance guarantees.

$\mathcal{R}_r$ (Figure 1, Right): We require that $\hat{\mathbf{X}}_t^{(i)} \subset C^{(i)}$, which is a new cvxMF constraint for both the classical and online setting. Theoretical guarantees for the underlying algorithm follow from small and fairly-obvious modifications in the proof for the $\mathcal{R}_u$ case, assuming error-free sample classification.

## 3 Online Algorithm

The proposed online cvxMF method for solving $\mathcal{R}_u$ consists of two procedures, described in Algorithms 1 and 2. Algorithm 1 describes the initialization of the main procedure in Algorithm 2. Algorithm 1 generates an initial estimate for the basis $\mathbf{D}_0$ and for the representative regions $\{\mathrm{cvx}(\hat{\mathbf{X}}_0^{(i)})\}_{i \in [k]}$. A similar initialization was used in classical cvxMF, with the bases vectors obtained either through clustering (on a potentially subsampled dataset) or through random selection and additional processing [2]. During initialization, one first collects a fixed prescribed number of $N$ data samples, summarized in $\hat{\mathbf{X}}$. Subsequently, one runs the K-means algorithm on the collected samples to obtain a clustering, described by the cluster indicator matrix $\mathbf{H} \in \{0, 1\}^{N \times k}$, in which $\mathbf{H}[n, i] = 1$ if the $n$-th sample lies in cluster $i$. The sizes of the generated clusters $\{N_i\}_{i \in [k]}$ are used as fixed cardinalities of the representative sets of the online methods. The initial estimate of the basis $\mathbf{D}_0[i]$ equals the average of the samples inside the cluster, i.e. $\mathbf{D}_0 \triangleq \hat{\mathbf{X}} \mathbf{H} \, \mathrm{diag}(1/N_1, \dots, 1/N_k)$.

Note again that initialization is performed using only a constant number of $N$ samples. Hence, K-means clustering does not significantly contribute to the complexity of the online algorithm. Second, to ensure that the restricted online cvxMF algorithm instantiates each cluster with at least one data sample, one needs to take into account the size of the smallest cluster (discussed in the Supplement).

---

**Algorithm 1** Initialization

1: **Input:** i.i.d samples $\underline{x}_1, \underline{x}_2, \dots, \underline{x}_N$ of a random vector $\underline{X} \in \mathbb{R}^m$ summarized in $\hat{\mathbf{X}}$.
2: Run K-means on $\hat{\mathbf{X}}$ to generate the cluster indicator matrix $\mathbf{H} \in \{0, 1\}^{N \times k}$ and determine the initial cluster sizes (subsequent representative set sizes) $N_i, i \in [k]$.
3: Compute $\mathbf{D}_0$ and $\hat{\mathbf{X}}_0^{(i)} \in \mathbb{R}^{m \times N_i}, \ \forall i \in [k]$, according to:

$$\mathbf{D}_0 = \hat{\mathbf{X}} \mathbf{H} \, \mathrm{diag}(1/N_1, \dots, 1/N_k)$$

and summarize the initial representative sets of the clusters into matrices $\hat{\mathbf{X}}_0^{(i)}, i = [k]$.
4: **Return:** $\mathbf{D}_0, \{\hat{\mathbf{X}}_0^{(i)}\}_{i \in [k]}$.

---

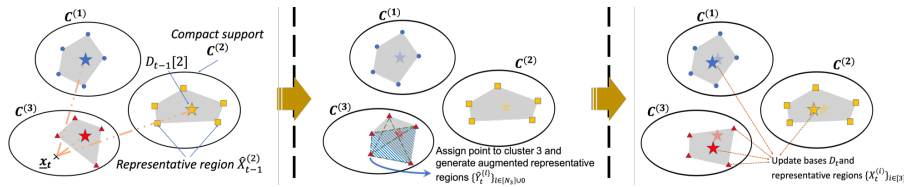

Figure 2: Illustration of one step of the online cvxMF algorithm with multiple-representative regions.

Following initialization, Algorithm 2 sequentially selects one sample $\underline{x}_t$ at a time and then updates the current representative sets $\hat{\mathbf{X}}_t^{(i)}, i \in [k]$, and bases $\mathbf{D}_t$. More precisely, after computing the coefficient vector $\underline{\alpha}_t$ in Step 5, one places the sample $\underline{x}_t$ into the appropriate cluster, indexed by $i_t$. The $N_{i_t}$-subsets

---

**Algorithm 2** Online cvxMF

---

1: **Input:** Data samples $\underline{x}_t$, a parameter $\lambda \in \mathbb{R}$, and the maximum number of iterations $T$.
2: **Initialization:** Compute $\mathbf{D}_0$, $\{\hat{\mathbf{X}}_0^{(i)}\}_{i \in [k]}$ using Algorithm 1. Set $\mathbf{A}_0 = \mathbf{0}$, $\mathbf{B}_0 = \mathbf{0}$.
3: **for** $t = 1$ to $T$ **do**
4:    Sample $\underline{x}_t$ from $\underline{\mathbf{X}}$.
5:    Update $\underline{\alpha}_t$ according to:

$$\underline{\alpha}_t = \arg\min_{\underline{\alpha} \in \mathbb{R}^k} \frac{1}{2} \left\| \underline{x}_t - \mathbf{D}_{t-1}\underline{\alpha} \right\|_2^2 + \lambda \|\underline{\alpha}\|_1. \tag{5}$$

6:    Set $\mathbf{A}_t = \frac{1}{t}\left((t-1)\mathbf{A}_{t-1} + \underline{\alpha}_t\underline{\alpha}_t^T\right)$   and   $\mathbf{B}_t = \frac{1}{t}\left((t-1)\mathbf{B}_{t-1} + \underline{x}_t\,\underline{\alpha}_t^T\right)$.
7:    Choose the index of the basis $i_t$ to be updated according to $i_t = \text{Uniform}([k])$.
8:    Generate the augmented representative regions $\left\{\hat{\mathbf{Y}}_t^{\{l\}}\right\}_{l \in [N_{i_t}] \cup \{0\}}$:

$$\hat{\mathbf{Y}}_t^{\{0\}} = \hat{\mathbf{X}}_{t-1}^{(i_t)}$$

$$\left\{\hat{\mathbf{Y}}_t^{\{l\}}\right\}_{l \in [N_{i_t}]} : \quad \hat{\mathbf{Y}}_t^{\{l\}}[j] = \begin{cases} \hat{\mathbf{X}}_{t-1}^{(i_t)}[j], & \text{if } j \in [N_i] \setminus l \\ \underline{x}_t, & \text{if } j = l. \end{cases} \tag{6}$$

9:    Update $\{\hat{\mathbf{X}}_t^{(i)}\}_{i \in [k]}$ and $\mathbf{D}_t$ by executing the following two steps:
   a.    Compute $l^\star, \hat{\mathbf{D}}^\star$ by solving the optimization problems:

$$l^\star, \hat{\mathbf{D}}^\star = \underset{\substack{l, \mathbf{D} \text{ s.t.} \\ \mathbf{D}[j] \in \text{cvx}\left(\hat{\mathbf{X}}_{t-1}^{(j)}\right) j \neq i_t, \\ \mathbf{D}[i_t] \in \text{cvx}\left(\hat{\mathbf{Y}}_t^{\{l\}}\right)}}{\arg\min} \quad \frac{1}{t}\sum_{n=1}^{t}\left(\frac{1}{2}\left\|\underline{x}_n - \mathbf{D}\underline{\alpha}_n\right\|_2^2 + \lambda\|\underline{\alpha}_n\|_1\right),$$

$$= \underset{\substack{l, \mathbf{D} \text{ s.t.} \\ \mathbf{D}[j] \in \text{cvx}\left(\hat{\mathbf{X}}_{t-1}^{(j)}\right) j \neq i_t, \\ \mathbf{D}[i_t] \in \text{cvx}\left(\hat{\mathbf{Y}}_t^{\{l\}}\right)}}{\arg\min} \quad \frac{1}{2}\text{Tr}(\mathbf{D}^T\mathbf{D}\mathbf{A}_t) - \text{Tr}(\mathbf{D}^T\mathbf{B}_t). \tag{7}$$

   b.    Set

$$\hat{\mathbf{X}}_t^{(i)} = \begin{cases} \hat{\mathbf{Y}}_t^{\{l^\star\}}, & \text{if } i = i_t \\ \hat{\mathbf{X}}_{t-1}^{(i)}, & \text{if } i \in [k] \setminus i_t, \end{cases}$$

$$\mathbf{D}_t = \hat{\mathbf{D}}^\star.$$

10: **end for**
11: **return** $\mathbf{D}_T$, the learned convex dictionary.

---

of $\{\text{col}(\hat{\mathbf{X}}_t^{(i_t)}) \cup \underline{x}_t\}$ (referred to as the augmented representative sets $\hat{\mathbf{Y}}_t^{\{l\}}$, $l \in [N_{i_t}] \cup \{0\}$), are used in Steps 8 and 9 to determine the new representative region $\hat{\mathbf{X}}_{t+1}^{(i_t)}$ for cluster $i_t$. To find the optimal index $l \in [N_{i_t}] \cup \{0\}$ and the corresponding updated basis $\mathbf{D}[i_t]$, in Step 9 we solve $N_{i_t}$ convex problems. The minimum of the optimal solutions of these optimization problems determines the new bases $\mathbf{D}_t$ and the representative regions $\hat{\mathbf{X}}_t^{(i)}$ (see Figure 2 for clarifications). Note that the combinatorial search step is executed on a constant-sized set of samples and is hence computationally efficient.

In Step 7, the new sample may be assigned to a cluster in two different ways. For the case $\mathcal{R}_u$, we use a random assignment. For the case $\mathcal{R}_r$, we need to perform the correct sample assignment in order to establish theoretical guarantees for the algorithm. Extensive simulations show that using $i_t = \arg\max \underline{\alpha}_t$ works very well in practice. Note that in either case, in order to minimize $g(\mathbf{D})$, one does not necessarily require an error-free classification process.

# 4   Convergence Analysis

In what follows, we show that the sequence of dictionaries $\{\mathbf{D}_t\}_t$ converges almost surely to a stationary point of $g(\mathbf{D})$ under assumptions similar to those used in [1], listed below.

(A.1) **The data distribution on a compact support set $C$ has bounded "skewness"**. The compact support assumption naturally arises in many practical applications. The bounded skewness assumption for the distribution of $\underline{X}$ reads as

$$\mathbb{P}(\|\underline{X} - \underline{p}\|_2 \leq r \mid \underline{X} \in C) \geq \kappa \operatorname{vol}\left(B(r, \underline{p})\right)/\operatorname{vol}(C), \tag{8}$$

where $C \triangleq \operatorname{cvx}(\bigcup_i C^{(i)})$, $\kappa$ is a positive constant and $B(r, \underline{p}) = \{\underline{x} : \|\underline{x} - \underline{p}\|_2 \leq r\}$ stands for the ball of radius $r$ around $\underline{p} \in C$. This assumption is satisfied for appropriate values of $\kappa$ and distributions of $\underline{X}$ that are "close" to uniform.

(A.2) **The quadratic surrogate functions $\hat{g}_t$ are strictly convex, and have Hessians that are lower-bounded by a positive constant $\kappa_1 > 0$**. It is straightforward to enforce this assumption by adding a term $\frac{\kappa_1}{2}\|\mathbf{D}\|_2^2$ to the surrogate or original objective function; this leads to replacing the positive semi-definite matrix $\frac{1}{t}\mathbf{A}_t$ in Equation (7) by $\frac{1}{t}\mathbf{A}_t + \kappa_1 I$.

(A.3) **The approximation-error function $\ell(x, \mathbf{D})$ is "well-behaved"**. We assume that the function $\ell(x, \mathbf{D})$ defined in Equation (3) is continuously differentiable, and that its expectation $g(\mathbf{D}) = \mathbb{E}_{\underline{X}}[\ell(\underline{X}, \mathbf{D})]$ is continuously differentiable and Lipschitz on the compact set $C$. This assumption parallels the one made in [1, Proposition 2], and it holds if the solution to Equation (3) is unique. The uniqueness condition can be enforced by adding a regularization term $\kappa\|\underline{\alpha}\|_2^2$ ($\kappa > 0$) to $\ell(\cdot)$ in Equation (3). This term makes the (LARS) optimization problem in Equation (5) strictly convex and hence ensures that it has a unique solution.

In addition, recall the definition of $\mathbf{D}_t$ and define $\mathbf{D}_t^\star$ as the global optima of the surrogate $\hat{g}_t(\mathbf{D})$,

$$\mathbf{D}_t = \underset{\mathbf{D}[i] \in \operatorname{cvx}(\hat{\mathbf{X}}_t^{(i)}),\, i \in [k]}{\arg\min} \hat{g}_t(\mathbf{D}),$$

$$\mathbf{D}_t^\star = \underset{\mathbf{D}[i] \in C,\, i \in [k]}{\arg\min} \hat{g}_t(\mathbf{D}).$$

## 4.1   Main Results

**Theorem 1.** *Under assumptions (A.1) to (A.3), the sequence $\{\mathbf{D}_t\}_t$ converges almost surely to a stationary point of $g(\mathbf{D})$.*

Lemma 2 bounds the difference of the surrogates for two different dictionary arguments. Lemma 3 establishes that restricting the optima of the surrogate function $\hat{g}_t(\mathbf{D})$ to the representative region $\operatorname{cvx}(\hat{\mathbf{X}}_t^{(i)})$ does not affect convergence to the asymptotic global optima $\mathbf{D}_\infty^\star$. Lemma 4 establishes that Algorithm 2 converges almost surely and that the limit is an optima $\mathbf{D}_\infty^\star$. Based on the results in Lemma 4, Theorem 1 establishes that the generated sequence of dictionaries $\mathbf{D}_t$ converges to a stationary point of $g(\mathbf{D})$. The proofs are relegated to the Supplement, but sketched below.

Let $\Delta_t \triangleq |\hat{g}_t(\mathbf{D}_t) - \hat{g}_t(\mathbf{D}_t^\star)|$ denote the difference between the surrogate functions for an unrestricted basis and a basis for which one requires $\mathbf{D}_t[i] \in \operatorname{cvx}\left(\hat{\mathbf{X}}_t^{(i)}\right)$. Then, one can show that

$$\Delta_t \leq \min\left\{\Delta_{t-1}, \left|\hat{g}_{t-1}(\mathbf{D}_t) - \hat{g}_{t-1}(\mathbf{D}_{t-1}^\star)\right|\right\} + O\left(\frac{1}{t}\right).$$

Based on an upper bound on the error of random coordinate descent used for minimizing the surrogate function and assumption (A.1), one can derive a recurrence relation for $\mathbb{E}[\Delta_t]$, described in the lemma below. This recurrence establishes a rate of decrease of $O\left(\frac{1}{t^{2/(m+2)}}\right)$ for $\mathbb{E}[\Delta_t]$.

**Lemma 2.** *Let $\Delta_t \triangleq \hat{g}_t(\mathbf{D}_t) - \hat{g}_t(\mathbf{D}_t^\star)$. Then*

$$\mathbb{E}[\Delta_t] \leq O\left(\frac{1}{t}\right) + \mathbb{E}[\Delta_{t-1}] - V_m \, \mathbb{E}[\Delta_{t-1}]^{\frac{m+2}{2}},$$

where $V_m \triangleq \frac{8\kappa \min_i p_i}{c_{\hat{g}}(m+2)\,\mathrm{vol}(C)} \frac{\pi^{\frac{m}{2}}}{\Gamma(\frac{m}{2}+1)} \left(\frac{2}{A\,k}\right)^{m/2}$, and $\kappa$ is the same constant used in Equation (8) of assumption (A.1). Also, $A = \max_i \mathbf{A}_t[i,i]$, $\forall\, t$, while $c_{\hat{g}}$ denotes a bound on the condition number of $\mathbf{A}_t$, $\forall t$, and $p_i$ denotes the probability of choosing $i_t = i$ in Step 7 of Algorithm 2.

Lemma 3 establishes that the optima $\mathbf{D}_t$ confined to the representative region and the global optima $\mathbf{D}_t^\star$ are close. From the Lipschitz continuity of $\hat{g}_t(\mathbf{D})$ asserted in assumptions (A.1) and (A.2), we can show that $\Delta_t = \|\mathbf{D}_t - \mathbf{D}_t^\star\|_{\hat{g}_t} \leq \|\mathbf{D}_t - \mathbf{D}_t^\star\|_2$. Lemma 3 then follows from Lemma 2 by applying the quasi-martingale convergence theorem stated in the Supplement.

**Lemma 3.** $\sum_t \frac{\|\mathbf{D}_t - \mathbf{D}_t^\star\|_2}{t+1}$ *converges almost surely.*

**Lemma 4.** *The following claims hold true:*

*P1)* $\hat{g}_t(\mathbf{D}_t)$ *and* $\hat{g}_t(\mathbf{D}_t^\star)$ *converge almost surely;*

*P2)* $\hat{g}_t(\mathbf{D}_t) - \hat{g}_t(\mathbf{D}_t^\star)$ *converges almost surely to 0;*

*P3)* $\hat{g}_t(\mathbf{D}_t^\star) - g(\mathbf{D}_t^\star)$ *converges almost surely to 0;*

*P4)* $g(\mathbf{D}_t^\star)$ *converges almost surely .*

The proofs of P1) and P2) involve completely new analytic approaches described in the Supplement.

## 5  Experimental Validation

We compare the approximation error and running time of our proposed online cvxMF algorithm with non-negative MF (NMF), cvxMF [2] and online MF [1]. For datasets with a ground truth, we also report the clustering accuracy. The datasets used include a) clusters of synthetic data samples; b) MNIST handwritten digits [32]; c) single-cell RNA sequencing datasets [31] and d) four other real world datasets from the UCI Machine Learning repository [30]. The largest sample size scales as $10^6$.

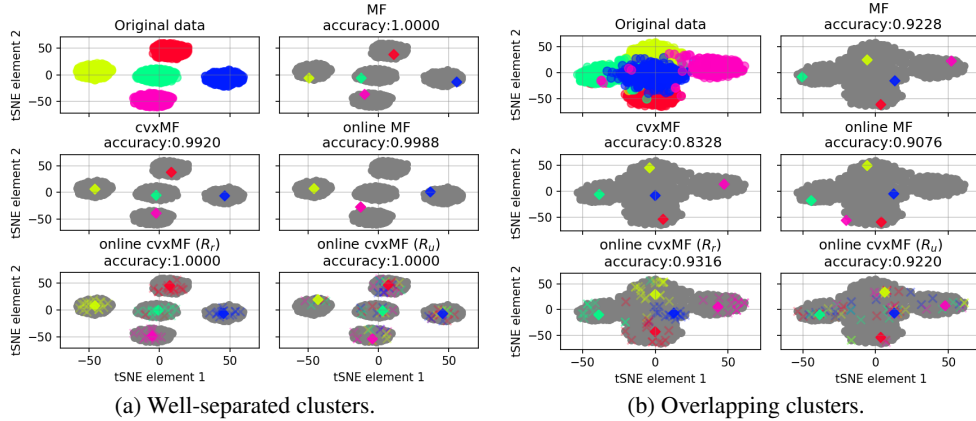

Figure 3: Results for Gaussian mixtures with color-coded clusters. Here, tSNE stands for the t-distributed stochastic neighbor embedding [33], in which the $x$-axis represents the first and the $y$-axis the second element of the embedding. Color-coded circles represent samples, diamonds represent basis vectors learned by the different algorithms, while crosses describe samples in the representative regions. The "interpretability property" can be easily observed visually.

**Synthetic Datasets.** The synthetic datasets were generated by sampling from a $3\sigma$-truncated Gaussian mixture model with 5 components, and with samples-sizes in $[10^3, 10^6]$. Each component Gaussian has an expected value drawn uniformly at random from $[0, 20]$ while the mixture covariance matrix equals the identity matrix $I$ ("well-separated clusters") or $2.5\,I$ ("overlapping clusters"). We ran the online cvxMF algorithm with both unconstrained $\mathcal{R}_u$ and restricted $\mathcal{R}_r$ representative regions, and

used the normalization factor $\lambda = c/\sqrt{m}$ suggested in [34]. After performing cross validation on an evaluation set of size 1000, we selected $c = 0.2$. Figure 3 shows the results for two synthetic datasets each of size $n = 2,500$ and with $N = 150$. The sample size was restricted for ease of visualization and to accommodate the cvxMF method which cannot run on larger sets. The number of iterations was limited to $1,200$. Both the cvxMF and online cvxMF algorithms generate bases that provide excellent representations of the data clusters. The MF and online MF method produce bases that are hard to interpret and fail to cover all clusters. Note that for the unrestricted version of cvxMF, samples of one representative set may belong to multiple clusters.

For the same Gaussian mixture model but larger datasets, we present running times and times to convergence (or, if convergence is slow, the maximum number of iterations) in Figure 4 (a) and (b), respectively. For well-separated synthetic datasets, we let $n$ increase from $10^2$ to $10^6$ and plot the results in (a). The non-online cvxMF algorithm becomes intractable after $10^4$ sample, while the cvxMF and MF easily scale for $10^6$ and more samples. To illustrate the convergence, we used a synthetic dataset with $n = 5,000$ in order to ensure that all four algorithms converge within 100s. Figure 4 (b) plots the approximation error $l_2 = \frac{1}{n}\|\mathbf{X} - \mathbf{D\Lambda}\|_2$ with respect to the running time. We chose a small value of $n$ so as to be able to run all algorithms, and for this case the online algorithms may have larger errors. But as already pointed out, as $n$ increases, non-online algorithms become intractable while the online counterparts operate efficiently (and with provable guarantees).

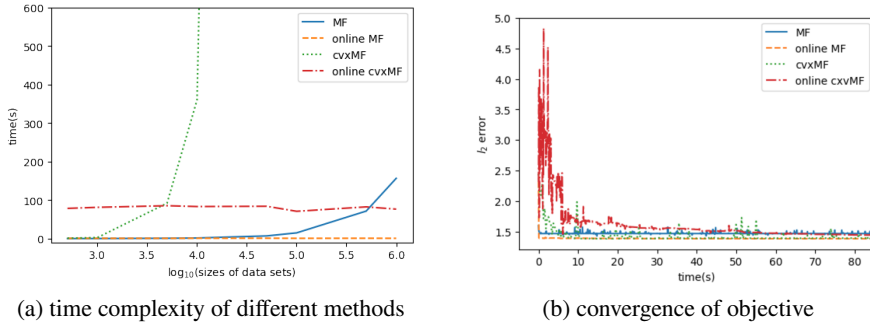

(a) time complexity of different methods      (b) convergence of objective

Figure 4: (a): Running times (s) vs. the log of the dataset sizes; (b) Running times (s) vs. the $l_2$ error.

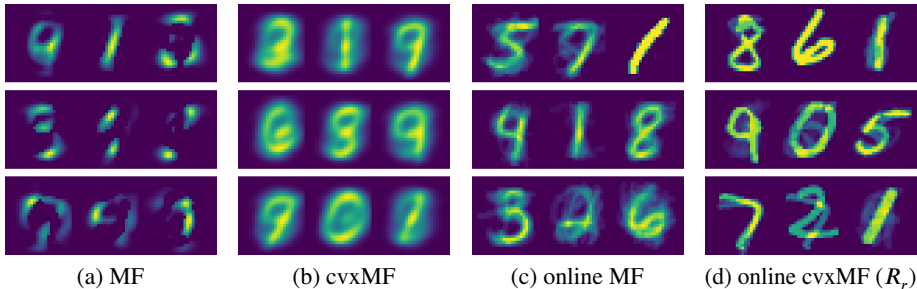

(a) MF      (b) cvxMF      (c) online MF      (d) online cvxMF ($R_r$)

Figure 5: MNIST results (as the eigenimage set is overcomplete, clustering accuracy is omitted).

**The MNIST Dataset.** The MNIST dataset was subsampled to a smaller set of $10,000$ images of resolution $28\times28$ to illustrate the performance of both the cvxMF and online cvxMF methods on image datasets. All algorithms ran $3,000$ iterations with $N = 150$ and $\lambda = 0.1$ to generate "eigenimages," capturing the characteristic features used as bases [35]. Figure 5 plots the first 9 eigenimages. The results for the $\mathcal{R}_u$ algorithm are similar to that of the non-online cvxMF algorithm and omitted. CvxMF produces blurry images since one averages all samples. The results are significantly better for the $\mathcal{R}_r$ case, as one only averages a small subset of representative samples.

**Single-Cell (sc) RNA Data.** scRNA datatsets contain expressions (activities) of all genes in individual cells, and each cell represents one data sample. Cells from the same tissue under same cellular condition tend to cluster, and due to the fact that that the sampled tissue are known, the cell labels are

known a priori. This setting allows us to investigate the $\mathcal{R}_r$ version of the online cvxMF algorithm to identify "typical" samples. For our dataset, described in more detail in the Supplement, the two non-online method failed to converge and required significantly larger memory. Hence, we only present results for the online methods. Results pertaining to real world datasets from the UCI Machine

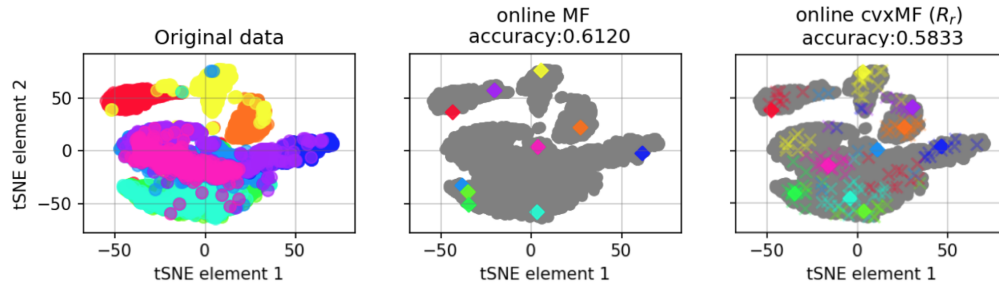

Figure 6: Results for the online methods executed on a blood-cell scRNA dataset.

Learning repository [30], also used for testing cvxMF [2], are presented in the Supplement.

## 6 Acknowledgement

The authors are grateful to Prof. Bruce Hajek for valuable discussions. This work was funded by the DB2K NIH 3U01CA198943-02S1, NSF/IUCR CCBGM Center, and the SVCF CZI 2018-182799 2018-182797.

## Footnotes

* Abhishek Agarwal and Jianhao Peng contribute equally to this work.

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
