[Supplementary Material]

# Supplement: Online Convex Matrix Factorization with Representative Regions

## S.1 Proofs of the Main Results

In what follows, we provide detailed proofs for the performance results pertaining to the online cvxMF algorithm presented in the main text. Technical auxiliary results needed to establish the findings in the text below are prefixed by A, and proved in Section S.2. Previously known results invoked in our discussion are presented in Section S.3.

We start with the proof of Lemma 2 from the main text, restated below for convenience.

**Lemma 2.** *Let* $\Delta_t \triangleq \hat{g}_t(\mathbf{D}_t) - \hat{g}_t(\mathbf{D}_t^\star)$. *Then*

$$\mathbb{E}\big[\Delta_t\big] \leq O\Big(\frac{1}{t}\Big) + \mathbb{E}\big[\Delta_{t-1}\big] - V_m \, \mathbb{E}\big[\Delta_{t-1}\big]^{\frac{m+2}{2}},$$

*where* $V_m \triangleq \frac{8\kappa \min_i p_i \frac{\pi^{\frac{m}{2}}}{\Gamma(\frac{m}{2}+1)}}{c_{\hat{g}}(m+2)\operatorname{vol}(C)}\Big(\frac{2}{A\,k}\Big)^{m/2}$, *and* $\kappa$ *is the same constant that appears in Equation* (8) *of assumption (A.1). Also,* $A = \max_i \mathbf{A}_t[i,i] \; \forall t$, *while* $c_{\hat{g}}$ *denotes an upper bound on the condition number of* $\mathbf{A}_t$, $\forall t$, *and* $p_i$ *denotes the probability of selecting* $i_t = i$ *in Step 7 of Algorithm 2.*

*Proof.* Let $\|\mathbf{D}_1 - \mathbf{D}_2\|_{\hat{g}} \triangleq |\hat{g}(\mathbf{D}_1) - \hat{g}(\mathbf{D}_2)|$. Then, we have the following inequalities

$$\Delta_t = \|\mathbf{D}_t - \mathbf{D}_t^\star\|_{\hat{g}_t} \tag{1a}$$

$$= \min\Big\{ \|\mathbf{D}_{t-1} - \mathbf{D}_t^\star\|_{\hat{g}_t}, \|\mathbf{D}_t - \mathbf{D}_t^\star\|_{\hat{g}_t} \Big\} \tag{1b}$$

$$\leq \min\{ \|\mathbf{D}_{t-1} - \mathbf{D}_{t-1}^\star\|_{\hat{g}_t} + \|\mathbf{D}_{t-1}^\star - \mathbf{D}_t^\star\|_{\hat{g}_t}, \ \|\mathbf{D}_t - \mathbf{D}_t^\star\|_{\hat{g}_t} \} \tag{1c}$$

$$\leq \min\Big\{ \|\mathbf{D}_{t-1} - \mathbf{D}_{t-1}^\star\|_{\hat{g}_t} + O\Big(\frac{1}{t}\Big), \ \|\mathbf{D}_t - \mathbf{D}_t^\star\|_{\hat{g}_t} \Big\}, \tag{1d}$$

where (1b) follows from Algorithm 2, since

$$\mathbf{D}_t = \underset{\mathbf{D}[i] \in \operatorname{cvx}(\hat{\mathbf{X}}_t^{(i)}) \, \bigcup \, \operatorname{cvx}(\hat{\mathbf{X}}_{t-1}^{(i)}), \, i \in [k]}{\arg\min} \hat{g}_t(\mathbf{D}),$$

and (1d) follows form the fact that $\hat{g}_t(\mathbf{D})$ is Lipschitz and from the bound in Proposition A4. Subsequently, from Proposition A4, we have $\|\mathbf{D}_{t-1} - \mathbf{D}_{t-1}^\star\|_{\hat{g}_t} = \|\mathbf{D}_{t-1} - \mathbf{D}_{t-1}^\star\|_{\hat{g}_{t-1}} + O\Big(\frac{1}{t}\Big)$ and $\|\mathbf{D}_t - \mathbf{D}_t^\star\|_{\hat{g}_t} \leq \|\mathbf{D}_t - \mathbf{D}_{t-1}^\star\|_{\hat{g}_{t-1}} + O\Big(\frac{1}{t}\Big)$. Therefore,

$$\Delta_t \leq \min\Big\{ \Delta_{t-1}, \|\mathbf{D}_t - \mathbf{D}_{t-1}^\star\|_{\hat{g}_{t-1}} \Big\} + O\Big(\frac{1}{t}\Big), \tag{2}$$

so that one has

$$\mathbb{E}\Big[\min\Big\{ \Delta_{t-1}, \|\mathbf{D}_t - \mathbf{D}_{t-1}^\star\|_{\hat{g}_{t-1}} \Big\} \,\Big|\, \mathcal{F}_{t-1}\Big]$$

$$\leq \Delta_{t-1} - \frac{2^{m/2+1}\kappa \upsilon_m}{(m+2)} \sum_{i \in [k]} \frac{p_i}{\mathbf{A}_{t-1}[i,i]^{m/2}\,\mathrm{vol}(C)} \left( \hat{g}_{t-1}(\mathbf{D}_{t-1}) - \min_{\substack{\underline{e}_i \in C \\ \underline{e}_j = \mathbf{D}_{t-1}[j], j \neq i}} \hat{g}_{t-1}([\underline{e}_1, \dots \underline{e}_k]) \right)^{m/2+1} \tag{3}$$

$$\leq \Delta_{t-1} - \frac{2^{m/2+1} p \kappa \upsilon_m}{(m+2)\,\mathrm{vol}(C) A^{m/2}} \sum_{i \in [k]} \left( \hat{g}_{t-1}(\mathbf{D}_{t-1}) - \min_{\substack{\underline{e}_i \in C \\ \underline{e}_j = \mathbf{D}_{t-1}[j], j \neq i}} \hat{g}_{t-1}([\underline{e}_1, \dots \underline{e}_k]) \right)^{m/2+1} \tag{4}$$

$$\leq \Delta_{t-1} - \frac{2 p \kappa \upsilon_m}{\mathrm{vol}(C)(m+2)} \left( \frac{2}{A\,k} \right)^{m/2} \left( \sum_{i \in [k]} \left( \hat{g}_{t-1}(\mathbf{D}_{t-1}) - \min_{\substack{\underline{e}_i \in C \\ \underline{e}_j = \mathbf{D}_{t-1}[j], j \neq i}} \hat{g}_{t-1}([\underline{e}_1, \dots \underline{e}_k]) \right) \right)^{m/2+1} \tag{5}$$

$$\leq \Delta_{t-1} - \frac{8 p \kappa \upsilon_m}{c_{\hat{g}}\,\mathrm{vol}(C)(m+2)} \left( \frac{2}{A\,k} \right)^{m/2} \left( \hat{g}_{t-1}(\mathbf{D}_{t-1}) - \min_{\mathbf{D}:\mathbf{D}[i] \in C \ \forall i} \hat{g}_{t-1}(\mathbf{D}) \right)^{,m/2+1} \tag{6}$$

where $p = \min_i p_i$, $A = \max_{t,i} \mathbf{A}_{t-1}[i,i]$, and $\upsilon_m = \frac{\pi^{\frac{m}{2}}}{\Gamma(\frac{m}{2}+1)}$, and Equations (3), (5) and (6) follow from Proposition A1, Hölder's inequality, and Proposition A2, respectively.

Using Equations (2) and (6) we have

$$\mathbb{E}\left[ \Delta_t \mid \mathcal{F}_{t-1} \right] \leq O\left( \frac{1}{t} \right) + \Delta_{t-1} - V_m\, \Delta_{t-1}^{\frac{m+2}{2}}$$
$$\implies \mathbb{E}\left[ \Delta_t \right] \leq O\left( \frac{1}{t} \right) + \mathbb{E}\left[ \Delta_{t-1} \right] - V_m\, \mathbb{E}\left[ \Delta_{t-1} \right]^{\frac{m+2}{2}},$$

which completes the proof. $\qquad\square$

We are now ready to prove Lemma 3 from the main text.

**Lemma 3.** $\sum_t \frac{\|\mathbf{D}_t - \mathbf{D}_t^\star\|}{t+1}$ converges almost surely.

*Proof.* Almost sure convergence of the sequence $\sum_{n \leq t} \frac{\|\mathbf{D}_n - \mathbf{D}_n^\star\|}{n}$ can be established using the quasi-martingale convergence Theorem A8. In this setting, the necessary condition in the convergence theorem is of the form

$$\sum_t \mathbb{E}\left[ \frac{\|\mathbf{D}_t - \mathbf{D}_t^\star\|}{t} \right] = O(1), \tag{7}$$

which follows from the sufficient condition, $\mathbb{E}[\|\mathbf{D}_t - \mathbf{D}_t^\star\|] = O\left( \frac{1}{t^{1/(m+2)}} \right)$. To prove this condition, we use Lemma 2 to compute the expected rate of decrease of $\Delta_t$. Then, from Lemma 2 and Proposition A3, we have

$$\mathbb{E}[\Delta_t] = O\left( \frac{1}{t^{2/(m+2)}} \right),$$
$$\implies \mathbb{E}[\|\mathbf{D}_t - \mathbf{D}_t^\star\|] \leq \sqrt{\mathbb{E}[\|\mathbf{D}_t - \mathbf{D}_t^\star\|^2]} \leq \sqrt{\frac{1}{\kappa_{\min}}\mathbb{E}[\Delta_t]} \tag{8}$$
$$\implies \mathbb{E}[\|\mathbf{D}_t - \mathbf{D}_t^\star\|] = O\left( \frac{1}{t^{1/(m+2)}} \right),$$

where Equation (8) follows by lower bounding the smallest eigenvalue of $\mathbf{A}_t$ $\forall t$ by $\kappa_{\min}$, based on assumption (**A**.2). This establishes the condition required in Equation (7) of Lemma 3. $\qquad\square$

We are now ready to prove Lemma 4 and Theorem 1 from the main text, restated below.

**Lemma 4.** *One has the following:*

*1.) $\hat{g}_t(\mathbf{D}_t)$ converges almost surely;*

*2.) $\hat{g}_t(\mathbf{D}_t^\star)$ converges almost surely;*

*3.) $\hat{g}_t(\mathbf{D}_t) - \hat{g}_t(\mathbf{D}_t^\star)$ converges almost surely to 0;*

*4.) $\hat{g}_t(\mathbf{D}_t^\star) - g(\mathbf{D}_t^\star)$ converges almost surely to 0;*

*5.) $g(\mathbf{D}_t^\star)$ converges almost surely.*

*Proof.* To prove that $\hat{g}_t(\mathbf{D}_t)$ converges almost surely, we follow the outline of the approach described [1]. The first three claims require new proof techniques and arguments, while the latter two results may be established using arguments similar to those described in [1].

Let $\gamma_t \triangleq \hat{g}_t(\mathbf{D}_t)$. For Claims 1 and 2, we prove almost sure convergence by showing that the positive stochastic process $\{\gamma_t\}$ satisfies

$$\sum_t \mathbb{E}[\mathbb{E}[\gamma_{t+1} - \gamma_t \mid \mathcal{F}_t]^+] < \infty, \text{ and} \tag{9}$$

$$\sum_t \mathbb{E}[\mathbb{E}[\gamma_{t+1}^\star - \gamma_t^\star \mid \mathcal{F}_t]^+] < \infty, \tag{10}$$

where $\mathcal{F}_t$ denotes the filtration up to time $t$. The above condition guarantees that the process is a quasi-martingale [2] that converges almost surely.

To establish the inequality on $\gamma_t$ needed in Equation (9), we write

$$\begin{aligned}
\gamma_{t+1} - \gamma_t &= \hat{g}_{t+1}(\mathbf{D}_{t+1}) - \hat{g}_t(\mathbf{D}_t) \\
&= \hat{g}_{t+1}(\mathbf{D}_{t+1}) - \hat{g}_{t+1}(\mathbf{D}_t) + \hat{g}_{t+1}(\mathbf{D}_t) - \hat{g}_t(\mathbf{D}_t) \\
&= \hat{g}_{t+1}(\mathbf{D}_{t+1}) - \hat{g}_{t+1}(\mathbf{D}_t) + \frac{\ell(\underline{x}_{t+1}, \mathbf{D}_t) - g_t(\mathbf{D}_t)}{t+1} + \frac{g_t(\mathbf{D}_t) - \hat{g}_t(\mathbf{D}_t)}{t+1} \tag{11} \\
&\leq \frac{\ell(\underline{x}_{t+1}, \mathbf{D}_t) - g_t(\mathbf{D}_t)}{t+1}, \tag{12}
\end{aligned}$$

where Equations (11) and (12) follows since Algorithm 2 guarantees that

$$\begin{aligned}
(t+1)\hat{g}_{t+1}(D_{t+1}) - t\hat{g}_t(D_t) &= \ell(\underline{x}_{t+1}, \mathbf{D}_t), \\
\hat{g}_{t+1}(\mathbf{D}_{t+1}) - \hat{g}_{t+1}(\mathbf{D}_t) &\leq 0, \\
g_t(\mathbf{D}_t) - \hat{g}_t(\mathbf{D}_t) &\leq 0.
\end{aligned}$$

Therefore, we have

$$\begin{aligned}
\mathbb{E}[\gamma_{t+1} - \gamma_t \mid \mathcal{F}_t] &\leq \frac{\mathbb{E}[\ell(\underline{x}_{t+1}, \mathbf{D}_t) \mid \mathcal{F}_t] - g_t(\mathbf{D}_t)}{t+1} \\
&= \frac{g(\mathbf{D}_t) - g_t(\mathbf{D}_t)}{t+1} \leq \frac{\|g - g_t\|_\infty}{t+1}.
\end{aligned}$$

The differences $g - g_t$ are bounded and smooth, which allows us to use Donsker's theorem [3, Lemma 19.36]. Hence,

$$\mathbb{E}\left[\mathbb{E}\left[\frac{\|g - g_t\|_\infty}{t+1}\right]^+\right] = O\left(\frac{1}{t^{3/2}}\right). \tag{13}$$

To prove Claim 2, we again use the quasi-martingale convergence theorem by establishing that Equation (10) holds. Let $\tilde{\ell}(\underline{x}, \mathbf{D}, \underline{\alpha}) \triangleq \frac{1}{2}\|\underline{x} - \mathbf{D}\underline{\alpha}\|^2 + \lambda\|\underline{\alpha}\|_1$ and note that $\tilde{\ell}(\underline{x}_{t+1}, \mathbf{D}_t, \underline{\alpha}_{t+1}) = \ell(\underline{x}_{t+1}, \mathbf{D}_t)$. Then, we have,

$$\gamma_{t+1}^\star - \gamma_t^\star = \hat{g}_{t+1}(\mathbf{D}_{t+1}^\star) - \hat{g}_t(\mathbf{D}_t^\star)$$

$$= \underbrace{\hat{g}_{t+1}(\mathbf{D}_{t+1}^{\star}) - \hat{g}_{t+1}(\mathbf{D}_t^{\star})}_{\leq 0} + \hat{g}_{t+1}(\mathbf{D}_t^{\star}) - \hat{g}_t(\mathbf{D}_t^{\star})$$

$$\leq \hat{g}_{t+1}(\mathbf{D}_t^{\star}) - \hat{g}_t(\mathbf{D}_t^{\star})$$

$$= \frac{\tilde{\ell}(\underline{x}_{t+1}, \mathbf{D}_t^{\star}, \underline{\alpha}_{t+1}) - \tilde{\ell}(\underline{x}_{t+1}, \mathbf{D}_t, \underline{\alpha}_{t+1})}{t+1} + \frac{\tilde{\ell}(\underline{x}_{t+1}, \mathbf{D}_t, \underline{\alpha}_{t+1}) - \hat{g}_t(\mathbf{D}_t^{\star})}{t+1}$$

$$\leq O(1)\frac{\|\mathbf{D}_t - \mathbf{D}_t^{\star}\|}{t+1} + \frac{\ell(\underline{x}_{t+1}, \mathbf{D}_t) - \hat{g}_t(\mathbf{D}_t^{\star})}{t+1}.$$

Therefore,

$$\mathbb{E}\left[\gamma_{t+1}^{\star} - \gamma_t^{\star} \mid \mathcal{F}_t\right] \leq O(1)\frac{\|\mathbf{D}_t - \mathbf{D}_t^{\star}\|}{t+1} + \frac{\mathbb{E}\left[\ell(\underline{x}_{t+1}, \mathbf{D}_t) \mid \mathcal{F}_t\right] - \hat{g}_t(\mathbf{D}_t^{\star})}{t+1}$$

$$= O(1)\frac{\|\mathbf{D}_t - \mathbf{D}_t^{\star}\|}{t+1} + \frac{g(\mathbf{D}_t) - \hat{g}_t(\mathbf{D}_t^{\star})}{t+1}$$

$$= O(1)\frac{\|\mathbf{D}_t - \mathbf{D}_t^{\star}\|}{t+1} + \frac{g(\mathbf{D}_t) - g_t(\mathbf{D}_t)}{t+1} + \frac{g_t(\mathbf{D}_t) - g_t(\mathbf{D}_t^{\star})}{t+1} + \underbrace{\frac{g_t(\mathbf{D}_t^{\star}) - \hat{g}_t(\mathbf{D}_t^{\star})}{t+1}}_{\leq 0}$$

$$\leq O(1)\frac{\|\mathbf{D}_t - \mathbf{D}_t^{\star}\|}{t+1} + \frac{g(\mathbf{D}_t) - g_t(\mathbf{D}_t)}{t+1}$$

$$\leq O(1)\frac{\|\mathbf{D}_t - \mathbf{D}_t^{\star}\|}{t+1} + \frac{\|g - g_t\|_{\infty}}{t+1}.$$

Thus, using Lemma 3 and Equation (13) we can establish Equation (10) similarly to Claim 1.

The proof of Claim 3 follows from Claims 1 and 2 if we show that $\hat{g}_t(\mathbf{D}_t) - \hat{g}_t(\mathbf{D}_t^{\star})$ converges to 0 in probability. We prove the latter by using the following well-known fact: If $\gamma_t$ converges almost surely, for all $\varepsilon, \delta \geq 0$ there exists a $T' \in \mathbb{N}$ such that

$$\mathbb{P}(\sup_{t \geq T} |\gamma_t - \gamma_T| \geq \varepsilon) \leq \delta, \ \forall T \geq T'. \tag{14}$$

For some constant $\kappa_1$, from Proposition A4 we have

$$|\gamma_{t+k} - \gamma_t^{\star}| = |\hat{g}_{t+k}(\mathbf{D}_{t+k}) - \hat{g}_t(\mathbf{D}_t^{\star})|$$

$$\leq \|\mathbf{D}_{t+k} - \mathbf{D}_t^{\star}\|_{\hat{g}_t} + \kappa_1\frac{k}{t}.$$

Let $p_i$ denote the probability that $i_t = i$ in Step 7 of Algorithm 2. Using Equation (8) in assumption (A.1), and the Lipschitz property of $\hat{g}_t$, we have

$$\mathbb{P}\left(\|\mathbf{D}_{t+k} - \mathbf{D}_t^{\star}\|_{\hat{g}_t} \leq \varepsilon \mid \mathcal{F}_t\right) \geq \prod_{i \in [k]}\left(p_i \kappa \frac{\text{vol}\left(B\left(\frac{\varepsilon}{\kappa_2}\right)\right)}{\text{vol}(\mathcal{C})}\right),$$

for some constant $\kappa_2$. Therefore, for $T \geq \max\left\{T', \frac{2k\,\kappa_1}{\varepsilon}\right\}$ we have

$$\prod_{i \in [k]}\left(p_i \kappa \frac{\text{vol}\left(B\left(\frac{\varepsilon}{2\kappa_2}\right)\right)}{\text{vol}(\mathcal{C})}\right) \leq \mathbb{P}\left(\|\mathbf{D}_{T+k} - \mathbf{D}_T^{\star}\|_{\hat{g}_T} \leq \frac{\varepsilon}{2} \mid |\gamma_T - \gamma_T^{\star}| \geq 2\varepsilon\right)$$

$$\leq \mathbb{P}\left(|\gamma_{T+k} - \gamma_T^{\star}| \leq \varepsilon \mid |\gamma_T - \gamma_T^{\star}| \geq 2\varepsilon\right)$$

$$\leq \mathbb{P}\left(|\gamma_{T+k} - \gamma_T| \geq \varepsilon \mid |\gamma_T - \gamma_T^{\star}| \geq 2\varepsilon\right)$$

$$\leq \mathbb{P}\left(\sup_{t \geq T}|\gamma_t - \gamma_T| \geq \varepsilon \mid |\gamma_T - \gamma_T^{\star}| \geq 2\varepsilon\right). \tag{15}$$

Combining Equation (14) with Equation (15), $\forall T \geq \max\left\{T', \frac{2k\,\kappa_1}{\varepsilon}\right\}$ we have

$$\delta \geq \mathbb{P}\left(\sup_{t \geq T}|\gamma_t - \gamma_T| \geq \varepsilon\right)$$

$$\geq \mathbb{P}\left(\sup_{t \geq T}|\gamma_t - \gamma_T| \geq \varepsilon \,\bigcap\, \left|\gamma_T - \gamma_T^\star\right| \geq 2\varepsilon\right)$$

$$= \mathbb{P}\left(\sup_{t \geq T}|\gamma_t - \gamma_T| \geq \varepsilon \,\bigg|\, \left|\gamma_T - \gamma_T^\star\right| \geq 2\varepsilon\right) \mathbb{P}\left(\left|\gamma_T - \gamma_T^\star\right| \geq 2\varepsilon\right)$$

$$\geq \prod_{i \in [k]}\left(p_i \kappa \frac{\mathrm{vol}\left(B\left(\frac{\varepsilon}{2\kappa_2}\right)\right)}{\mathrm{vol}(\mathcal{C})}\right) \mathbb{P}\left(|\gamma_T - \gamma_T^\star| \geq 2\varepsilon\right).$$

Hence, $|\gamma_T - \gamma_T^\star| \xrightarrow{\mathrm{P}} 0 \implies \hat{g}_t(\mathbf{D}_t) - \hat{g}_t(\mathbf{D}_t^\star) \xrightarrow{\mathrm{P}} 0$. Consequently, since $\hat{g}_t(\mathbf{D}_t)$ and $\hat{g}_t(\mathbf{D}_t^\star)$ converge almost surely, we have

$$\hat{g}_t(\mathbf{D}_t) - \hat{g}_t(\mathbf{D}_t^\star) \xrightarrow{\text{a.s.}} 0.$$

To prove Claims 4 and 5, we use the quasi-martingale convergence theorem of [2] and Equation (11) which gives us

$$\sum_t |\mathbb{E}[\gamma_{t+1} - \gamma_t \mid \mathcal{F}_t]| < \infty \implies \sum_t \frac{\hat{g}_t(\mathbf{D}_t) - g_t(\mathbf{D}_t)}{t+1} < \infty.$$

Therefore,

$$\infty > \sum_t \frac{\hat{g}_t(\mathbf{D}_t) - g_t(\mathbf{D}_t)}{t+1}$$

$$\geq \sum_t \frac{\hat{g}_t(\mathbf{D}_t^\star) - g_t(\mathbf{D}_t^\star)}{t+1} + \sum_t \frac{g_t(\mathbf{D}_t^\star) - g_t(\mathbf{D}_t)}{t+1}.$$

Using Lemma 3 and the Lipschitz continuity of $g_t(\mathbf{D})$, we have

$$\sum_t \frac{|g_t(\mathbf{D}_t^\star) - g_t(\mathbf{D}_t)|}{t+1} \leq \sum_t \frac{\kappa_2 \|\mathbf{D}_t^\star - \mathbf{D}_t\|}{t+1} < \infty$$

$$\implies \sum_t \frac{\hat{g}_t(\mathbf{D}_t^\star) - g_t(\mathbf{D}_t^\star)}{t+1} < \infty.$$

Therefore, since $\mathbf{D}_t^\star - \mathbf{D}_{t+1}^\star = O(1/t)$ and $\hat{g}_t(\mathbf{D}) - g_t(\mathbf{D})$ has a Lipschitz constant independent on $t$, using Lemma A5 we get

$$\hat{g}_t(\mathbf{D}_t^\star) - g_t(\mathbf{D}_t^\star) \xrightarrow{\text{a.s.}} 0.$$

The proofs of Claims 4 and 5 then follow from the Glivenko-Cantelli theorem [3, Thm 19.4] and Lemma 3. $\square$

**Theorem 1.** *Under assumptions (A.1) to (A.3), the sequence $\{\mathbf{D}_t\}_t$ converges almost surely to a stationary point of $g(\mathbf{D})$.*

*Proof.* The proof is similar to that of [1, Proposition 4] and follows from Algorithm 2 and Lemma 4. $\square$

## S.2 Auxiliary Proofs

This section contains the proofs of a number of auxiliary results used in the previous section.

Propositions A1 and A2 allow us to compute the expectation $\mathbb{E}[\Delta_{t+1}]$ and describe it recursively in terms of $\mathbb{E}[\Delta_t]$ in Lemma 2. While Proposition A1 follows from assumptions (A.1) to (A.3) and some simple algebra, proving Proposition A2 requires using properties of the random coordinate descent approach described in Algorithm 2.

**Proposition A1.** *It holds*

$$\mathbb{E}\left[\min\left\{\hat{g}_t(\mathbf{D}_t), \hat{g}_t(\mathbf{D}_{t+1})\right\} \,\Big|\, \mathcal{F}_t, \, i_{t+1} = i\right] \leq \hat{g}_t(\mathbf{D}_t) - V_m^{(i)}\left(\hat{g}_t(\mathbf{D}_t) - \min_{\substack{\mathbf{E}: \, \mathbf{E}[i]\in C \\ \mathbf{E}[j]=\mathbf{D}_t[j], j\neq i}} \hat{g}_t(\mathbf{E})\right)^{m/2+1},$$

*where* $V_m^{(i)} = \frac{2^{m/2+1}\kappa \frac{\pi^{\frac{m}{2}}}{\Gamma(\frac{m}{2}+1)}}{\mathbf{A}_t[i,i]^{m/2}(m+2)\,\mathrm{vol}(C)}$.

*Proof.* For ease of explanation, we prove the result for the case of two clusters only, i. e. for $k = 2$. It is straightforward to generalize the proof for larger values of $k$. Also, without loss of generality, we prove the inequality in Proposition A1 for $i = 1$ only. The proof for $i \in [k]$ follows along the same lines.

We rewrite the surrogate function $\hat{g}_t(\mathbf{D})$) as

$$\hat{g}_t(\mathbf{D}) = \frac{1}{2}\left(\left\|\mathrm{vec}(\mathbf{D}) - \frac{1}{\sqrt{\mathbf{I}_m \otimes \mathbf{A}_t}}\mathrm{vec}(\mathbf{B}_t)\right\|_{\mathbf{I}_m\otimes\mathbf{A}_t}^2 - \left\|\frac{1}{\sqrt{\mathbf{I}_m \otimes \mathbf{A}_t}}\mathrm{vec}(\mathbf{B}_t)\right\|_{\mathbf{I}_m\otimes\mathbf{A}_t}^2\right) + \eta_t, \quad (16)$$

where $\eta_t \triangleq \frac{1}{2t}\sum_{n\leq t}\left(\left\|\underline{x}_n\right\|^2 + 2\lambda\left\|\underline{\alpha}_n\right\|^2\right)$, $\mathbf{A}_t$ and $\mathbf{B}_t$ are matrices defined in Step 6 in Algorithm 2, $\|\underline{x}\|_{\mathbf{A}} \triangleq \sqrt{\underline{x}^T\mathbf{A}\underline{x}}$, $\mathbf{I}_m$ denotes the $m \times m$ identity matrix and $\otimes$ denotes the Kronecker product.

Let $\underline{c}_{t,1} \triangleq \frac{\mathbf{B}_t[1]-\mathbf{A}_t[1,2]\,\mathbf{D}_t[2]}{\mathbf{A}_t[1,1]}$ and $\underline{c}_{t,1}^{\star} \triangleq \arg\min_{\underline{e}_1\in C}\hat{g}_t([\underline{e}_1, \mathbf{D}_t[2]])$. Then, from Equation (16) we have

$$\hat{g}_t\left([\underline{d}_1, \mathbf{D}_t[2]]\right) = \frac{\mathbf{A}_t[1,1]}{2}\left\|\underline{d}_1 - \underline{c}_{t,1}\right\|^2 + a_t^{(1)}, \quad (17)$$

where $a_t^{(1)} \triangleq \frac{1}{2}\mathbf{A}_t[2,2]\|\mathbf{D}_t[2]\|^2 - \langle\mathbf{B}_t[2], \mathbf{D}_t[2]\rangle - \frac{\mathbf{A}_t[1,1]}{2}\left\|\underline{c}_{t,1}\right\|^2 + \eta_t$. For $\mathbf{U}_t$ defined as

$$\mathbf{U}_t : \mathbf{U}_t[l] \triangleq \begin{cases} \underline{x}_t & l = i_t \\ \mathbf{D}_{t-1}[l] & l \neq i_t, \end{cases} \quad (18)$$

we have based on Equation (17)

$$\mathbb{E}\left[\min\left\{\hat{g}_t(\mathbf{D}_t), \hat{g}_t(\mathbf{D}_{t+1})\right\} \,\Big|\, \mathcal{F}_t, \, i_{t+1} = 1\right]$$

$$\leq \mathbb{E}\left[\min\left\{\hat{g}_t(\mathbf{D}_t), \hat{g}_t(\mathbf{U}_{t+1})\right\} \,\Big|\, \mathcal{F}_t, \, i_{t+1} = 1\right]$$

$$= a_t^{(1)} + \frac{\mathbf{A}_t[1,1]}{2}\mathbb{E}_{\underline{X}}\left[\min\left\{\left\|\mathbf{D}_t[1] - \underline{c}_{t,1}\right\|^2, \left\|\underline{X} - \underline{c}_{t,1}\right\|^2\right\}\right]$$

$$\leq a_t^{(1)} + \frac{\mathbf{A}_t[1,1]}{2}\mathbb{E}_{\underline{X}}\left[\min\left\{\left\|\mathbf{D}_t[1] - \underline{c}_{t,1}\right\|^2, \left\|\underline{X} - \underline{c}_{t,1}^{\star}\right\|^2 + \left\|\underline{c}_{t,1}^{\star} - \underline{c}_{t,1}\right\|^2\right\}\right]$$

$$= a_t^{(1)} + \frac{\mathbf{A}_t[1,1]}{2}\left\|\underline{c}_{t,1}^{\star} - \underline{c}_{t,1}\right\|^2 + \frac{\mathbf{A}_t[1,1]}{2}\mathbb{E}_{\underline{X}}\left[\min\left\{\left\|\mathbf{D}_t[1] - \underline{c}_{t,1}\right\|^2 - \left\|\underline{c}_{t,1}^{\star} - \underline{c}_{t,1}\right\|^2, \left\|\underline{X} - \underline{c}_{t,1}^{\star}\right\|^2\right\}\right]$$

$$\leq a_t^{(1)} + \frac{\mathbf{A}_t[1,1]}{2}\left\|\mathbf{D}_t[1] - \underline{c}_{t,1}\right\|^2 - \frac{\mathbf{A}_t[1,1]\kappa\upsilon_m}{(m+2)\,\mathrm{vol}(C)}\left(\left\|\mathbf{D}_t[1] - \underline{c}_{t,1}\right\|^2 - \left\|\underline{c}_{t,1} - \underline{c}_1^{\star}\right\|^2\right)^{m/2+1} \quad (19)$$

$$= \hat{g}_t(\mathbf{D}_t) - \frac{2^{m/2+1}\kappa\upsilon_m}{\mathbf{A}_t[1,1]^{m/2}(m+2)\,\mathrm{vol}(C)}\left(\hat{g}_t(\mathbf{D}_t) - \min_{\substack{\underline{e}_1\in C \\ \underline{e}_j=\mathbf{D}_t[j],j\neq 1}}\hat{g}_t([\underline{e}_1, \cdots \underline{e}_k])\right)^{m/2+1}, \quad (20)$$

where Equation (20) follows since $a_t^{(1)} + \frac{1}{2}\mathbf{A}_t[1,1]\left\|\underline{c}_{t,1}^\star - \underline{c}_{t,1}\right\|^2 = \min_{\underline{e}_1 \in C} \hat{g}([\underline{e}_1, \mathbf{D}[2]])$. The proof of Equation (19) is given below.

For any real value $r$, a point $\underline{p}_1 \in C$, and the norm $\left\|\underline{X} - \underline{p}_1\right\| \triangleq U_1$, one has

$$\mathbb{E}_{\underline{X}}\left[\min\{r^2, U_1^2\}\right]$$

$$= \left(1 - \mathbb{P}(U_1 \leq r)\right)r^2 + \mathbb{P}(U_1 \leq r)\int_0^{r^2}\left(1 - \mathbb{P}\left(U_1 \leq \sqrt{s} \mid U_1 \leq r\right)\right)ds$$

$$= r^2 - \int_0^{r^2}\mathbb{P}(U_1 \leq \sqrt{s})\,ds$$

$$\leq r^2 - \frac{\kappa v_m}{\text{vol}(C)}\int_0^{r^2}s^{m/2}\,ds \tag{21}$$

$$= r^2 - \frac{2\kappa v_m}{(m+2)\,\text{vol}(C)}r^{m+2},$$

where $v_m = \frac{\pi^{\frac{m}{2}}}{\Gamma(\frac{m}{2}+1)}$ and Equation (21) follows from assumption (**A.1**). □

**Proposition A2.** *Let $\mathcal{D} \triangleq \{\mathbf{D} : \mathbf{D}[i] \in C, \forall i\}$. Then,*

$$\left(\hat{g}_t(\mathbf{D}_t) - \min_{\mathbf{D} \in \mathcal{D}}\hat{g}_t(\mathbf{D})\right) \leq (4/c_{\hat{g}})\sum_{i=1}^k\left(\hat{g}_t(\mathbf{D}_t) - \min_{\substack{\mathbf{E}: \\ \mathbf{E}[i]\in C \\ \mathbf{E}[j]=\mathbf{D}_t[j], j\neq i}}\hat{g}_t(\mathbf{E})\right)$$

*where $c_{\hat{g}}$ denotes an upper bound on the condition numbers of the matrices $\mathbf{A}_t$, $\forall t$.*

*Proof.* Let $b_l$ and $b_u$ denote upper and lower bounds (respectively) on the eigenvalues of $\mathbf{A}_t$, $\forall t$. Let

$$\mathbf{D}_t^{(i)*} = \underset{\substack{\mathbf{D} \text{ s.t. } \mathbf{D}[i]\in C \\ \mathbf{D}[j]=\mathbf{D}_t[j], j\neq i}}{\arg\min}\hat{g}_t(\mathbf{D})$$

and

$$\mathbf{D}_t^\star = \underset{\mathbf{D}\in\mathcal{D}}{\arg\min}\,\hat{g}_t(\mathbf{D})$$

and

$$h_t(\mathbf{D}) \triangleq \hat{g}_t(\mathbf{D}_t^\star) + \frac{b_u}{2}\left\|\mathbf{D} - \mathbf{D}_t^\star\right\|^2.$$

Note that $h_t(\mathbf{D})$ is a strongly convex function, and that $h_t(\mathbf{D}_t^\star) = \hat{g}_t(\mathbf{D}_t^\star)$, $\nabla h_t(\mathbf{D}) = b_u(\mathbf{D} - \mathbf{D}_t^\star)$. Thus, from the definition of $\hat{g}_t(\mathbf{D})$ in Equation (16), we have

$$\hat{g}_t(\mathbf{D}_t) - \hat{g}_t(\mathbf{D}_t^\star) = \frac{1}{2}\left(\left\|\text{vec}(\mathbf{D}_t) - \frac{1}{\sqrt{\mathbf{I}_m \otimes \mathbf{A}_t}}\text{vec}(\mathbf{B}_t)\right\|_{\mathbf{I}_m\otimes\mathbf{A}_t}^2 - \left\|\text{vec}(\mathbf{D}_t^\star) - \frac{1}{\sqrt{\mathbf{I}_m \otimes \mathbf{A}_t}}\text{vec}(\mathbf{B}_t)\right\|_{\mathbf{I}_m\otimes\mathbf{A}_t}^2\right)$$

$$\leq \frac{1}{2}\left\|\text{vec}(\mathbf{D}_t) - \text{vec}(\mathbf{D}_t^\star)\right\|_{\mathbf{I}_m\otimes\mathbf{A}_t}^2 \leq \frac{b_u}{2}\left\|\mathbf{D}_t - \mathbf{D}_t^\star\right\|^2$$

$$= \frac{1}{2b_u}\|\nabla h_t(\mathbf{D}_t)\|^2. \tag{22}$$

Next, define a sequence $\{\mathbf{D}_{t,j}\}_{j=0}^\infty \subset \mathcal{D}$ such $\mathbf{D}_{t,0} = \mathbf{D}_t$ and $\mathbf{D}_{t,j+1} = \Pi_{\mathcal{D}}(\mathbf{D}_{t,j} - \frac{1}{b_l}\nabla\hat{g}_t(\mathbf{D}_{t,j}))$, where $\mathbf{D}_{t,0} = \mathbf{D}_t$ and $\Pi_{\mathcal{D}}(\mathbf{X})$ denotes the projection of $\mathbf{X}$ onto $\mathcal{D}$ (i.e., the sequence is the result of successive iterations of the projected gradient descent algorithm (PGD) for solving $\arg\min_{\mathbf{D}\in\mathcal{D}}\hat{g}_t(\mathbf{D})$).

From the convergence guarantees for PGD, we know that we can asymptotically reach the optimal $\mathbf{D}_t$, i.e., $\mathbf{D}_{t,\infty} = \mathbf{D}_t^\star$. Let $\widetilde{\nabla}\hat{g}_t(\mathbf{D}) \triangleq b_l\left(\mathbf{D} - \Pi_D\left(\mathbf{D} - \frac{1}{b_l}\nabla\hat{g}_t(\mathbf{D})\right)\right)$. Then, the update rule for the previously introduced sequence may be rewritten as

$$\mathbf{D}_{t,j+1} = \mathbf{D}_{t,j} - \frac{1}{b_l}\widetilde{\nabla}\hat{g}_t(\mathbf{D}_{t,j}).$$

Therefore,

$$\|\mathbf{D}_{t,j+1} - \mathbf{D}_t^\star\|^2 = \left\|\mathbf{D}_{t,j} - \frac{1}{b_l}\widetilde{\nabla}\hat{g}_t(\mathbf{D}_{t,j}) - \mathbf{D}^\star\right\|^2$$

$$= \|\mathbf{D}_{t,j} - \mathbf{D}^\star\|^2 + \frac{1}{b_l^2}\|\widetilde{\nabla}\hat{g}_t(\mathbf{D}_{t,j})\|^2 - \frac{2}{b_l}\left\langle\widetilde{\nabla}\hat{g}_t(\mathbf{D}_{t,j}), \mathbf{D}_{t,j} - \mathbf{D}^\star\right\rangle$$

$$\leq \|\mathbf{D}_{t,j} - \mathbf{D}^\star\|^2 + \frac{1}{b_l^2}\|\widetilde{\nabla}\hat{g}_t(\mathbf{D}_{t,j})\|^2 - \frac{1}{b_l}\left(b_u\|\mathbf{D}_{t,j} - \mathbf{D}^\star\|^2 + \frac{1}{b_l}\|\widetilde{\nabla}\hat{g}_t(\mathbf{D}_{t,j})\|^2\right)$$

(23)

$$= \left(1 - \frac{b_u}{b_l}\right)\|\mathbf{D}_{t,j} - \mathbf{D}^\star\|^2$$

$$\implies \|\mathbf{D}_{t,j+1} - \mathbf{D}_t^\star\| \leq \left(1 - \frac{b_u}{2b_l}\right)\|\mathbf{D}_{t,j} - \mathbf{D}_t^\star\|$$

$$\implies \|\mathbf{D}_{t,j} - \mathbf{D}_t^\star\| \leq \frac{2b_l}{b_u}\left(\|\mathbf{D}_{t,j} - \mathbf{D}_t^\star\| - \|\mathbf{D}_{t,j+1} - \mathbf{D}_t^\star\|\right)$$

$$\leq \frac{2b_l}{b_u}\|\mathbf{D}_{t,j} - \mathbf{D}_{t,j+1}\|,$$

where Equation (23) follows from [4, Section 2.2]. Hence, we have

$$\left\|\nabla h_t(\mathbf{D}_{t,j})\right\|^2 = b_u^2\left\|\mathbf{D}_{t,j} - \mathbf{D}_t^\star\right\|^2 \leq 4b_l^2\left\|\mathbf{D}_{t,j} - \mathbf{D}_{t,j+1}\right\|^2 = 4\left\|\widetilde{\nabla}\hat{g}_t(\mathbf{D}_{t,j})\right\|^2. \tag{24}$$

Next, we define coordinate PGD sequences $\{\mathbf{D}_{t,j}^{(i)}\}_{j=0}^{j=\infty}$ for all clusters $i \in [k]$ similarly as was done in the preceding discussion, with $\mathbf{D}_{t,0}^{(i)} = \mathbf{D}_t$ and $\mathbf{D}_{t,\infty}^{(i)} = \mathbf{D}_t^{(i)*}$. Let $\widetilde{\nabla}\hat{g}_t(\mathbf{D})$ and $\nabla\hat{g}_t(\mathbf{D})$ stand for the $l^{\text{th}}$ columns of the gradients $\widetilde{\nabla}\hat{g}_t(\mathbf{D})$ and $\nabla\hat{g}_t(\mathbf{D})$, respectively. Then, the update rule for each cluster reads as

$$\mathbf{D}_{t,j+1}^{(i)} = \mathbf{D}_{t,j}^{(i)} - \frac{1}{b_l}\widetilde{\nabla}^{(i)}\hat{g}_t\left(\mathbf{D}_{t,j}^{(i)}\right),$$

where $\widetilde{\nabla}_l^{(i)}\hat{g}_t(\mathbf{D})$ denotes the $l^{\text{th}}$ column of the gradient $\widetilde{\nabla}^{(i)}\hat{g}_t(\mathbf{D})$, defined as

$$\widetilde{\nabla}_l^{(i)}\hat{g}_t(\mathbf{D}) \triangleq \begin{cases} b_l\left(\mathbf{D}[i] - \Pi_{C^{(i)}}\left(\mathbf{D}[i] - \frac{1}{b_l}\nabla_i\hat{g}_t(\mathbf{D})\right)\right) & l = i, \\ \mathbf{0} & l \neq i. \end{cases}$$

Thus, $\widetilde{\nabla}_i^{(i)}\hat{g}_t(\mathbf{D}) = \widetilde{\nabla}_i\hat{g}_t(\mathbf{D})$. Hence,

$$\left\|\widetilde{\nabla}\hat{g}_t(\mathbf{D}_{t,j})\right\|^2 = \sum_{i=1}^k\left\|\widetilde{\nabla}_i\hat{g}_t(\mathbf{D}_{t,j})\right\|^2$$

$$\implies \left\|\widetilde{\nabla}\hat{g}_t(\mathbf{D}_{t,0})\right\|^2 = \sum_{i=1}^k\left\|\widetilde{\nabla}_i\hat{g}_t(\mathbf{D}_{t,0})\right\|^2 = \sum_{i=1}^k\left\|\widetilde{\nabla}_i^{(i)}\hat{g}_t\left(\mathbf{D}_{t,0}^{(i)}\right)\right\|^2$$

$$\leq \sum_{i=1}^k 2b_l\left(\hat{g}_t(\mathbf{D}_{t,0}^{(i)}) - \hat{g}_t(\mathbf{D}_{t,1}^{(i)})\right) \tag{25}$$

$$\leq 2b_l\sum_{i=1}^k\left(\hat{g}_t(\mathbf{D}_t) - \hat{g}_t(\mathbf{D}_t^{(i)*})\right), \tag{26}$$

where Equation (25) follows from the property of the PGD algorithm given in [4, Section 2.2] with respect to $\{\mathbf{D}_{t,j}^{(i)}\}_j$, since the function $\hat{g}_t([\mathbf{D}_t[1], \ldots, \mathbf{D}_t[i-1], \underline{e}_i, \mathbf{D}_t[i+1], \ldots, \mathbf{D}_t[k]])$ is $b_l$-smooth in $\underline{e}_i$, $\forall i$.

Next, combining Equations (22), (24) and (26), we obtain

$$\hat{g}_t(\mathbf{D}_t) - \hat{g}_t(\mathbf{D}_t^\star) \le \frac{4b_l}{b_u} \sum_{i=1}^{k} \left( \hat{g}_t(\mathbf{D}_t) - \hat{g}_t(\mathbf{D}_t^{(i)*}) \right).$$

$\square$

The next result, Proposition A3, establishes that the deterministic recursion in Lemma 2 generates a fast-converging sequence.

**Proposition A3.** *For any $A_1 \ge 0$, the recursion $0 \le A_{n+1} \le A_n - \kappa_1 (A_n)^m + \frac{\kappa_2}{n}$ implies that $A_n = O\left( \frac{1}{n^{1/m}} \right)$, for all possible $m \in (1, \infty)$.*

*Proof.* Let

$$F(n) \triangleq \left( \frac{\kappa_2 + \alpha}{\kappa_1 \, n} \right)^{1/m},$$

where $\alpha$ is the smallest positive value such that

$$\frac{\alpha}{n} \ge \left( \frac{\kappa_2 + \alpha}{\kappa_1} \right)^{1/m} \left( \left( \frac{1}{n} \right)^{1/m} - \left( \frac{1}{n} \right)^{1/m} \right), \ \forall n \ge 1.$$

Then, we use inductive arguments to establish that for some constant integers $N_0 < n_0$, one has

$$A_n \in [0, F(n - N_0)], \ \forall n \ge n_0. \tag{27}$$

To do so, we first prove that $A_n \xrightarrow{n} 0$. For large enough $n$ we have

$$\begin{aligned}
A_{n+1} &\le A_n - \kappa_1 \left( A_n \right)^m + \frac{\kappa_2}{n} \\
&\le \max_a \ a - \kappa_1 a^m + \frac{\kappa_2}{n} \\
&= \left( \frac{1}{\kappa_1 m} \right)^{1/m} \left( 1 - \frac{1}{m} \right) + \frac{\kappa_2}{n} \\
&\le \left( \frac{1}{\kappa_1 m} \right)^{1/m}.
\end{aligned}$$

Therefore, since $x - \kappa_1 x^m$ is monotonic for $x \le \left( \frac{1}{\kappa_1 m} \right)^{1/m}$, it is easy to see that $A_n \xrightarrow{n} 0$. Thus, there must exist integers $N_0 \le n_0$, and $A_{n_0} \le F(n_0 - N_0) \le \left( \frac{1}{\kappa_1 m} \right)^{1/m}$. Let us assume that $A_t \in [0, F(t - N_0)]$, $\forall t \le n$ for some $n > n_0$. Then, we need to show that $A_{n+1} \in [0, F(n + 1 - N_0)]$. We have

$$\begin{aligned}
A_{n+1} &\le A_n - \kappa_1 \left( A_n \right)^m + \frac{\kappa_2}{n} \\
&\le \max_{a \in [0, F(n - N_0)]} a - \kappa_1 a^m + \frac{\kappa_2}{n} \\
&= F(n - N_0) - \kappa_1 \left( F(n - N_0) \right)^m + \frac{\kappa_2}{n}.
\end{aligned}$$

Thus, proving $F(n - N_0) - \kappa_1 \left( F(n - N_0) \right)^m + \frac{\kappa_2}{n} \le F(n + 1 - N_0)$ will complete the induction. We observe that

$$F(n - N_0) - \kappa_1 \left( F(n - N_0) \right)^m + \frac{\kappa_2}{n} \le F(n + 1 - N_0)$$

$$\Longleftrightarrow \ F(n - N_0) - F(n + 1 - N_0) \leq \kappa_1 \big(F(n - N_0)\big)^m - \frac{\kappa_2}{n}. \tag{28}$$

The left hand side of Equation (28) satisfies

$$F(n - N_0) - F(n + 1 - N_0) = \left(\frac{\kappa_2 + \alpha}{\kappa_1}\right)^{1/m}\left(\left(\frac{1}{n - N_0}\right)^{1/m} - \left(\frac{1}{n + 1 - N_0}\right)^{1/m}\right)$$

$$\leq \frac{\alpha}{n - N_0},$$

due to the choice of $\alpha$. The right hand side of Equation (28) satisfies

$$\kappa_1\big(F(n - N_0)\big)^m - \frac{\kappa_2}{n} = -\frac{\kappa_2}{n} + \frac{\kappa_2 + \alpha}{n - N_0}$$

$$\geq \frac{\alpha}{n - N_0}.$$

This completes the proof. $\qquad\square$

Proposition A4 establishes a result similar to one proved in [1].

**Proposition A4.** *We have*

$$\|\mathbf{D}_{t+1}^\star - \mathbf{D}_t^\star\| = O\left(\frac{1}{t}\right). \tag{29}$$

*Furthermore, for any $\mathbf{D} \in \mathcal{D}$ and $t \geq 1$*

$$\hat{g}_t(\mathbf{D}) - \hat{g}_{t-1}(\mathbf{D}) = O\left(\frac{1}{t}\right). \tag{30}$$

*Proof.* The proof follows along the same lines as that of Lemma 1 in [1] and is hence omitted. $\quad\square$

## S.3   Collection of results used in the main proofs

**Lemma A5. [Positive Converging Sums].**
*Let $a_n$, $b_n$ be two real sequences such that for all $n$, $a_n \geq 0, b_n \geq 0, \sum_{n=1}^{\infty} a_n = \infty, \sum_{n=1}^{\infty} a_n b_n < \infty$, $\exists K > 0$ s.t. $|b_{n+1} - b_n| < K a_n$. Then, $\lim_{n \to +\infty} b_n = 0$.*

*Proof.* Suppose that $b_n \geq \varepsilon$ for all $n > N \in \mathbb{N}$ and any $\varepsilon > 0$. Then, $\sum_n a_n b_n$ converges to $\infty$. Therefore, for any $\varepsilon > 0$, $\exists$ indices $m_j, n_j$ such that

$$b_n > \varepsilon, \ \forall n \in [m_j, n_j)$$

$$b_n \leq \varepsilon, \ \forall n \in [n_j, m_{j+1}).$$

Since, $\sum_{n \geq 1} a_n b_n < \infty$ there exists an index $m_J$ such that $\sum_{n \geq m_j} a_n b_n \leq \frac{\varepsilon^2}{K}$. Let $n \in [m_j, n_j)$ for any $j \geq J$. Then,

$$|b_n - b_{n_j}| \leq \sum_{m=n}^{n_j - 1} |b_{n+1} - b_n|$$

$$\leq \sum_{m=n}^{n_j - 1} K a_n$$

$$\leq \frac{K}{\varepsilon} \sum_{m=n}^{n_j - 1} a_n b_n$$

$$\leq \frac{K}{\varepsilon} \sum_{m \geq n} a_n b_n$$

$$\leq \frac{K}{\varepsilon} \frac{\varepsilon^2}{K} = \varepsilon.$$

Therefore, $b_n \leq b_{n_j} + \varepsilon \leq 2\varepsilon$ for all $n \in [m_j, n_j)$ and for any $j \geq J$. Thus, $b_n \leq 2\varepsilon$ for all $n \geq m_J$. $\quad\square$

**Theorem A6. Donsker's theorem [3, Ch 19.2, lemma 19.36]**
*Let $F = \{f_\theta : \chi \to \mathbb{R}, \theta \in \Theta\}$ be a set of measurable functions indexed by elements of a bounded subset $\Theta$ of $\mathbb{R}^d$. Suppose that there exists a constant $K$ such that*

$$|f_{\theta_1}(x) - f_{\theta_2}(x)| \leq K\|\theta_1 - \theta_2\|_2,$$

*for all $\theta_1, \theta_2 \in \Theta$ and $x \in \chi$.*

*Then, for i.i.d random variables $X_1, X_2, \ldots$ and for any $f$ in $F$, define $\mathbb{P}_n f$, $\mathbb{P}f$ and $\mathbb{G}_n f$ as*

$$\mathbb{P}_n f = \frac{1}{n}\sum_{i=1}^{n} f(X_i),$$

$$\mathbb{P}f = \mathbb{E}_X[f(X)],$$

$$\mathbb{G}_n f = \sqrt{n}(\mathbb{P}_n f - \mathbb{P}f).$$

*Assume further that for all $f$, $\mathbb{P}f^2 < \delta^2$ and $\|f\|_\infty < M$. Then,*

$$\mathbb{E}_P[\|\mathbb{G}_n\|_F] = O(1),$$

*where $\|\mathbb{G}_n\|_F = \sup_{f \in F}|\mathbb{G}_n f|$.*

**Lemma A7.** *Let $f_{\mathbf{D}}(\underline{x}) \triangleq \ell(\underline{x}, D)$. Then, for all $\mathbf{D}$, $f_{\mathbf{D}}(\underline{x})$ satisfies the necessary conditions for Donsker's theorem (Theorem A6)*

1. *$|f_{\mathbf{D}_1}(x) - f_{\mathbf{D}_2}(x)| \leq K\|\mathbf{D}_1 - \mathbf{D}_2\|_2, \ \forall \mathbf{D}_1, \mathbf{D}_2 \in \mathcal{C}$;*

2. *$\mathbb{P}f^2 < \delta^2, \forall f \in \{f_{\mathbf{D}} : \mathbf{D} \in \mathcal{C}\}$;*

3. *$\|f\|_\infty < M, \forall f \in \{f_{\mathbf{D}} : \mathbf{D} \in \mathcal{C}\}$;*

The proof follows from the definition of $\ell(\underline{x}, \mathbf{D})$ and Propositions 2 and 3 in [1].

**Theorem A8. Quasi-Martingale Convergence Theorem [2].**
*Let $(\Omega, \mathcal{F}, P)$ be a measurable probability space, let $\gamma_t$, $t \geq 0$ be a stochastic sequence and $\mathcal{F}_t$ its induced filtration. Let*

$$\delta_t = \begin{cases} 1 & \text{if } \mathbb{E}\left[\gamma_{t+1} - \gamma_t \mid \mathcal{F}_t\right] > 0, \\ 0 & \text{otherwise.} \end{cases}$$

*If for all $t$, $\gamma_t \geq 0$ and $\sum_{t=1}^{\infty} \mathbb{E}[\delta_t(\gamma_{t+1} - \gamma_t)] < \infty$, then $\gamma_t$ is a quasi-martingale and converges almost surely. Moreover,*

$$\sum_{t=1}^{\infty} |\mathbb{E}\left[\gamma_{t+1} - \gamma_t \mid \mathcal{F}_t\right]| < +\infty \ a.s.$$

## S.4 Experimental results of real world dataset

All experiments presented in this work were performed using software written in Python and executed on a Linux machine with an Intel Xeon Gold CPU @ 3.20GHz and with 376GBs of memory.

The scRNA dataset comprise 10 cell types and $94,655$ samples. The cell types and their numbers are listed below:

- B cells: $10,085$;
- Cd14 monocytes: $2,612$ (the smallest cluster size);
- Cd34 cells: $9,232$;
- Cd4 t helper cells: $11,213$;
- Cd56 natural killer cells: $8,385$;
- Cytotoxic T cells: $10,209$;
- Memory T cells: $10,224$;
- Naive cytotoxic cells: $11,953$ (the largest cluster size);

Table 1: Description of the data sets, their average clustering accuracy and average running times.

| | synthetic | iris | wine | iono-sphere | 20news | MNIST |
|---|---|---|---|---|---|---|
| # samples | 2500 | 150 | 178 | 351 | 2034 | 10000 |
| # clusters | 5 | 3 | 3 | 2 | 20 | 10 |
| $\hat{n}$ | 15 | 15 | 15 | 15 | 20 | 25 |
| average accuracy over 10 experiments | | | | | | |
| MF | 0.926 | 0.6667 | 0.6180 | 0.6410 | 0.5349 | 0.566 |
| cvxMF | 0.848 | 0.7667 | 0.6910 | 0.7179 | 0.5831 | 0.620 |
| online MF | 0.898 | 0.7040 | 0.6742 | 0.6803 | 0.5782 | 0.503 |
| online cvxMF | 0.899 | 0.6793 | 0.6573 | 0.6963 | 0.5683 | 0.543 |
| running time (second) on each dataset | | | | | | |
| cvxMF | 25 | 0.14 | 0.21 | 0.46 | 22.2 | 667 |
| online cvxMF | 38 | 6.4 | 7.2 | 6.4 | 58.4 | 120 |

- Naive T cells: $10, 479$;
- Regulatory T cells: $10, 263$.

Besides the synthetic datasets and MNIST dataset mentioned in the paper, we also tested our method on the Iris, Wine, Ionosphere and 20news datasets from the UCI Machine Learning repository [5]. A detailed description of these dataset can be found in the following Table 1. Since in this case all data samples have known labels, the performance of the algorithms may be evaluated through clustering accuracy. Clustering accuracy is calculated by first sorting the columns/rows of the confusion matrix of dimensions $k \times k$, capturing the label assignments, so as to maximize the trace. Subsequently, the trace is normalized by the number of samples to produce the desired accuracy values. The computational complexity is measured in terms of average running time for each algorithm to converge. All experiments were performed with $\lambda = 0.1$ and $1, 200$ iterations. The accuracy reported is the average over 10 experiments.

From the tabulated results, one can see that when compared to its non-online counterpart, online cvxMF does not incur a significant loss in accuracy (the loss is upper bounded by 6.7%). When the size of the dataset increases, the complexity of cvxMF increase dramatically (up to 6000 times). The online cvxMF algorithm is less affected by the increasing size of $n$, and the increase in complexity is due to increasing $\hat{n}$, the number of clusters $k$ and dimension $m$ of the data.