[Reviews · NeurIPS 2019]

Reviewer 1



After rebuttal: The authors addressed my concerns by explaining their choice not to include a discussion of them modified algorithm was motivated by the aim to keep the paper focused and concise. My recommendation remains accept, 9. The authors consider a version of the convex MF/dictionary learning problem appropriate for datasets with a cluster structure: they constrain the basis elements to be convex combinations of exemplar data points, where they explicitly attempt to enforce sparse combinations by using l1 penalties on the coefficients, and design the algorithm to select the exemplar data points appropriately from the clusters so that the basis elements lie within the convex hull of the individual clusters. The problem addressed by the paper is well-motivated, and the intuitions justifying the design of the algorithm are well-explained. The novelty lies in the way the authors use the clustering structure assumed to be present in the dataset to show that their algorithm converges to a stationary point of the convex MF objective. I did not read the proofs in the supplement, but it seems that the techniques used will be of interest to those designing similar non-convex stochastic online algorithms.

Reviewer 2



The authors propose an online version of convex matrix factorization (MF) which cannot be made online by adapting the seminal work of Mairal et al., 2000.They provide an unconstrained and a constrained version of online convex MF. The experiments show the advantages of convex MF over other MF versions as well as the online version of convex MF over the batch version. The technical parts of the paper could have been written more clearly for a wider audience but it should be ok for experts in the field. The experiments section is clearer and makes a strong demonstration for the proposed method. In general, there is room for improvement in the structuring.

Reviewer 3



Overall, this paper is well written. The authors have proven the effectiveness of the proposed algorithm through sufficient analysis. However, I have several concerns regarding the proposed algorithm. First question is that what is a reasonable choice of N in your initialization phase. You use N samples to construct the initial representative sets as well as the dictionary matrix. As an online algorithm, I think your convergence bound should include N as a parameter here. But I cannot see it from the theorem you presented in the paper. Can you explain it? The number of samples in each cluster is assumed to be constant over time. What is the intuition behind this? Does this mean in the initialization, you already have enough samples to expand the representation space? For the restricted online cvsMF algorithm, you assume that the new sample always gets to the correct cluster which I think is not possible in real world. Especially, you only have limit number of samples to initialize your K clusters. How could it be possible that you always get correct assignment?

[Author Response · NeurIPS 2019]

We thank all the reviewers for carefully reading of the manuscript and constructive comments. We addressed the issues raised in our response below, mostly focusing on the numerous questions raised by Reviewer 3.

**Reviewer #1: Assumptions A.2 and A.3 used in Algorithm** 2. Assumption A.2 was also used in Mairal et al. [1], while A.3 is a common assumption in linear regression analysis and it relates to the LARS problem in our online cvxMF setting. We agree with Reviewer #1 that the modifications described in detail on page 6 of the manuscript may have produced practical performance improvements in the proposed algorithm. We will report on these results in the revised paper. We wanted to avoid notational overload and exactly match our analysis with every step of the algorithm which is the reason why we did not dwell on testing the proposed modifications.

**Reviewer #2: Improving the clarity of the technical sections.** We agree with the reviewer that the notation may be hard to follow, but the problem setup is such that most of it is necessary. We will make every attempt to further improve readability through examples and detailed comments in the Supplement.

**Reviewer #3:** There seem to be several misunderstandings regarding the steps of our algorithm and its analysis.

**1. MF versus NMF.** Our results (both the proof and the algorithm) are independent of the non-negativity assumption and apply to both NMF and MF problems. To enforce non-negativity, several simple projection steps, akin to those described in Mairal et al. [1], suffice. For simplicity of exposition, and to be able to compare our results with the main result in [1], the derivations were presented for the classical MF problem only.

**2. The role of K-means in the online algorithm.** We would like to point out that K-means is used only once to initialize the representative sets and is not an intrinsic component of the online algorithm. Furthermore, K-means is performed on small subsampled datasets, as running it on complete datasets is time consuming and unnecessary.

**3. The role of the constant $N$.** A reasonable choice of $N$ in the initialization and update phase depends on the size of the dataset $n$, the dimensions of the data points $m$ and the number of clusters $k$. What is important to observe is that $N$ is kept constant throughout in order to reduce the storage footprint and to ensure low-complexity online processing. Whenever a new point is fetched, it is compared against other points for inclusion into the representative regions. To maintain the list constant, for every added point another point is removed. Also, the reviewer is correct in observing that $N$ does not feature in the convergence results, which are asymptotic and do not imply anything about the convergence rate. Clearly, if the point dimension $m$ is large, it is beneficial to increase $N$. A final remark is that the $N$ representative points are used to generate the convex bases for the space and not to "cover" the clusters.

**4. The perfect assignment assumption for the restricted online cvxMF is unrealistic.** First, we point out that both the algorithm and the convergence analysis in our paper mostly focus on the unrestricted online cvxMF problem. In this case, the cluster assignment for a newly retrieved data point is made uniformly at random over the set of all possible clusters; we proved that this random assignment suffices to find a stationary point of the problem. The set of representative points still contains $N$ data samples, and the convex hull property is still valid, except that the bases are not required to lie in the convex hull of points from the same cluster. Clearly, this unrestricted case can only have better performance than the restricted setting as nothing in the algorithm or the proof relies on correct classification. This point is illustrated in Figure 1: Even when the new sample (triangle in second cluster) is misclassified as belonging to the bottom cluster, the basis representing the latter cluster (red star) remains in the convex hull of representative set; the newly added point is retained in the representative set if the objective decreases and discarded otherwise.

The restricted version is proposed because of its practical utility and ease of interpretation as we explained in great detail on page 2, in the Introduction of the main text. In this setting, one requires the representative set to be partitioned into $k$ representative subsets, each of which is restricted to be contained in its corresponding cluster. The basis is consequently restricted to be in the convex hull of data points from the same cluster. To satisfy these conditions we indeed require a "perfect assignment," which

Figure 1: An incorrect assignment of a data point to a different cluster does not affect the "convex hull" constraint; the misclassified point (triangle) is still part of the representative set used to describe the basis (star). Hence, in this unrestricted representative region setting, classification errors do not influence the performance of the method nor do they compromise convergence guarantees.

is possible in many supervised and semi-supervised learning tasks where the labels of points are known beforehand. We once again point out that the main goal of our algorithm is not to accurately classify the data point, but to find the optimal convex bases in an online manner and retain a small list of representative points from the data set. Even with the availability of labels or perfect cluster assignment it is nontrivial to compute bases that satisfy the restricted convexity constraint with an online algorithm. As shown by our experiments, the heuristic classification step that we proposed to use in practice for the restricted version works very well and provides excellent results for several real world tasks.

[Meta-Review · NeurIPS 2019]

Matrix factorization is notoriously computational expensive. The authors an efficient online algorithm which can be shown to converge to a batch objective under certain strong assumptions. Solid experimental results and theoretical results are given. The reviewers are in consensus that this paper should be accepted.